# Conformalized Decision Risk Assessment

**Wenbin Zhou**[1]**, Agni Orfanoudaki**[2] **& Shixiang Zhu**[1]
[1]Carnegie Mellon University    [2]University of Oxford
`{wenbinz2, shixianz}@andrew.cmu.edu`
`agni.orfanoudaki@sbs.ox.ac.uk`

## Abstract

High-stakes decisions in healthcare, energy, and public policy have long depended on human expertise and heuristics, but are now increasingly supported by predictive and optimization-based tools. A prevailing paradigm in operations research is predict-then-optimize, where predictive models estimate uncertain inputs and optimization models recommend decisions. However, such approaches often sideline human judgment, creating a disconnect between algorithmic outputs and expert intuition that undermines trust and adoption in practice. To bridge this gap, we propose `CREDO`, a framework that, for any candidate decision proposed by human experts, provides a distribution-free upper bound on the probability of suboptimality—informed by both the optimization structure and the data distribution. By combining inverse optimization geometry with conformal generative prediction, `CREDO` delivers statistically rigorous risk certificates. This framework allows human decision-makers to audit and validate their decisions under uncertainty, strengthening the alignment between algorithmic tools and human intuition.

## 1 Introduction

Decision-making under uncertainty in domains with significant societal and economic consequences requires not only the identification of nominally optimal solutions, but also rigorous quantification of the probability that any candidate decision maintains optimality when uncertain parameters are realized (Zhu et al., 2022). The prevailing *predict-then-optimize* (PTO) paradigm addresses uncertainty by first estimating unknown parameters through predictive models (*e.g.*, future demand or patient outcomes), then solving an optimization problem to recommend actions (Bertsimas & Kallus, 2020; Elmachtoub & Grigas, 2022). This pipeline has become the foundation of many data-driven decision systems across a wide spectrum of applications (Bertsimas et al., 2021; Tian et al., 2023).

Despite its ubiquity, the PTO approach has two fundamental shortcomings in high-stakes contexts. First, these systems often function as black boxes that prescribe decisions without revealing their robustness or sensitivity to uncertainty. As a result, they do not provide any transparency as to whether alternative decisions might perform comparably or better under different parameter realizations. This opacity makes it difficult for human decision-makers to gauge confidence in algorithmic recommendations or determine when to override them with their own expertise (Li & Zhu, 2024; Zhang et al., 2025). Second, such pipelines are inherently limited to point predictions that cannot capture distributional complexity. For example, under multi-modal parameter distributions, these methods recommend decisions optimized for expected values that fall between modes, where the true parameters may have negligible probability (Sim et al., 2024). In practice, this means PTO may not only fail to guide human judgment effectively but also recommend harmful or misleading actions.

These shortcomings are especially problematic given that high-stakes real-world decision-making rarely relies solely on algorithmic prescriptions. Experienced practitioners often propose alternatives based on domain knowledge that extends beyond available data, such as insights about rare events, operational constraints, or risk factors not captured in historical records. Yet, current optimization frameworks offer no principled way to evaluate these expert-generated decisions, creating a disconnect between algorithmic tools and practitioner expertise. Decision-makers need methods to rigorously assess any candidate solution, enabling them to compare algorithmic prescriptions with alternatives derived from human judgment.

To bridge this gap, we propose the complementary *decide-then-assess* paradigm. Rather than replacing human expertise with model prescriptions, our goal is to support human judgment by auditing candidate decisions in a data driven manner. Specifically, we ask: *For a user-specified decision, how likely is it to remain optimal under the true, unknown realization of uncertainty?* This perspective enables rigorous risk assessment for any candidate decision, regardless of whether it originates from optimization algorithms, expert judgment, or external constraints.

Building on this principle, we introduce CREDO—Conformalized Risk Estimation for Decision Optimization—a framework that quantifies, for any candidate decision, a distribution-free upper bound on the probability of suboptimality (Figure 1). Unlike scenario-based planning, which evaluates decisions against a discrete set of possible futures, CREDO quantifies optimality probability over the entire parameter distribution with rigorous statistical guarantees. The framework rests on two key insights: $(i)$ in a broad class of optimization problems, the optimal solution is a deterministic function of the objective parameters (Chan et al., 2025), which allows us to invert this mapping and characterize the set of outcomes under which a decision remains optimal; and $(ii)$ by combining conformal prediction (Shafer & Vovk, 2008) with generative modeling, we can estimate the probability mass of this set, yielding valid, data-driven upper bounds on decision risk. The resulting tool provides informative and computationally efficient risk certificates, empowering human decision-makers to audit and validate their choices under uncertainty, while systematically exposing discrepancies between human intuition and empirical data.

Our contributions can be summarized as follows:

- Problem Formulation: We formalize decision risk assessment as computing distribution-free bounds on the probability that a candidate decision remains optimal under parameter uncertainty, establishing its connection to inverse optimization and conformal prediction.
- CREDO Framework: We propose CREDO, a distribution-free framework that upper bounds the probability that a given decision is suboptimal. We further develop an efficient closed-form estimator for linear programs that enables practical implementation in real-world decision contexts and establish marginal conservativeness guarantees on our risk estimates, characterizing the accuracy and variance properties of our estimator.
- Empirical Validation: Through controlled synthetic experiments and a real-world power grid planning application, we verify CREDO's theoretical guarantees and demonstrate its practical advantages in risk assessment

Figure 1: Illustrative example of the problem setting of CREDO. For a given candidate decision and relevant data, CREDO estimates the risk of the likelihood for the decision to be optimal.

accuracy, highlighting its calibration, efficiency, and ability to distinguish between robust and fragile decisions under uncertainty.

**Related Works** Decision-making under uncertainty is a fundamental challenge in operations research and machine learning. A large body of work addresses this through robust optimization (RO), which ensures decisions perform well under worst-case realizations within an uncertainty set (Bertsimas & Thiele, 2006). Distributionally robust optimization (DRO) generalizes this by accounting for ambiguity in the underlying distribution (Delage & Ye, 2010; Duchi & Namkoong, 2021; Levy et al., 2020; Rahimian & Mehrotra, 2022). More recently, the PTO paradigm has gained traction as a practical framework for data-driven decision-making (Bertsimas & Kallus, 2020; Elmachtoub et al., 2020; Lepenioti et al., 2020). It involves a two-stage procedure: first predicting unknown parameters via machine learning, then solving a deterministic optimization problem using these estimates. This has inspired decision-focused learning (DFL) approaches, where model training directly targets decision quality by differentiating through the optimization layer (Amos & Kolter, 2017; Chen et al., 2025; Mandi et al., 2024; Shah et al., 2022; Wang et al., 2025; Wilder et al., 2019). While these frameworks may account for risks through their modeling assumptions, they do not explicitly quantify the level of risk associated with each decision. Our work addresses this gap directly.

Methodologically, our work builds upon the literature on conformal prediction (CP), a general statistical framework that enables distribution-free uncertainty quantification by constructing calibrated

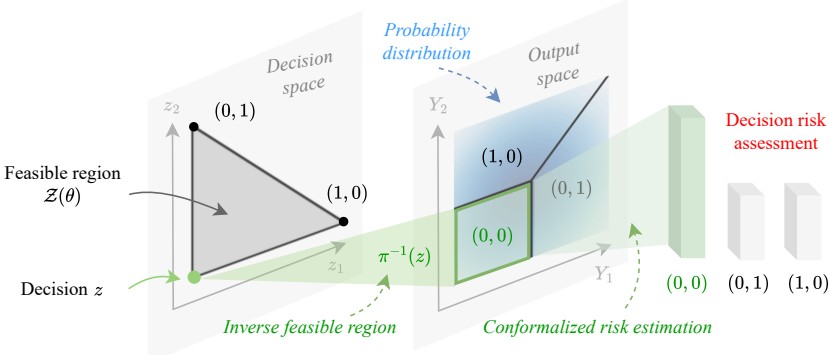

Figure 2: The overall architecture of the proposed framework. It contains two main steps: ($i$) Map the given candidate decision $z$ (*e.g.*, $(0.0)$) to its inverse feasible region $\pi^{-1}(z)$; ($ii$) Conservatively assess the risk via conformalized risk estimation over $\pi^{-1}(z)$ using collected data of $X$ and $Y$.

prediction sets under mild assumptions (Papadopoulos et al., 2002; Shafer & Vovk, 2008; Vovk et al., 2005). Two strands of recent research are particularly relevant to our approach. First, a growing body of work explores replacing traditional point prediction models in CP with generative models, enabling better modeling of stochastic outputs and yielding tighter prediction sets (Wang et al., 2023; Zheng & Zhu, 2024; Zhou et al., 2024). Our methodology adopts this generative modeling perspective as a key component to improve the accuracy of our risk estimates. A parallel line of research also investigates the "inverted" use of conformal prediction, where the goal is to estimate the miscoverage rate corresponding to a fixed prediction set (Prinster et al., 2022; 2023; Singh et al., 2024; Gauthier et al., 2025a). The technique we use is most similar to the concurrent work of (Gauthier et al., 2025a), leveraging recent advancements in e-value conformal prediction (Vovk, 2025; Balinsky & Balinsky, 2024; Gauthier et al., 2025b). However, our framework departs from their work by focusing on the decision risk assessment setting rather than being a pure conformal prediction task.

Integration of CP and optimization has also been explored in recent works for decision-making, including RO (Kiyani et al., 2025; Andrews & Chen, 2025; Patel et al., 2024; Chenreddy & Delage, 2024; Lin et al., 2024; Chan et al., 2024), DFL (Cortes-Gomez et al., 2024), and human-related contexts (Hullman et al., 2025). Specifically, Kiyani et al. (2025) shows that prediction sets (*e.g.*, via conformal prediction) naturally define robust optimization uncertainty sets whose miscoverage level encodes risk tolerance under VaR, Andrews & Chen (2025) demonstrate that ambiguity-averse decisions can be framed as minimax choices over confidence sets, yielding high-probability loss certificates. In comparison, our work differs in both role and objective: instead of using conformal sets as inputs to produce optimal decisions, we invert conformal prediction to quantify how likely a given decision is optimal, making our framework descriptive (auditing decisions) rather than prescriptive (recommending them), and thus complementary to these prior approaches.

Finally, our study contributes to a growing body of research on human-AI interaction in decision-making. Recent work has investigated how machine learning algorithms can serve as advisory tools to support human decisions (Chen et al.; Grand-Clément & Pauphilet, 2024; Hullman et al., 2025; Orfanoudaki et al., 2022). A parallel line of research has explored how generative models can be leveraged to present a more diverse set of decisions, thereby encouraging exploration and aiding the development of improved decision strategies (Ajay et al., 2022; Krishnamoorthy et al., 2023; Li & Zhu, 2024). Our work advances further in quantifying the risk associated with each candidate decision, enriching the informational quality of decision support—particularly when multiple dimensions of decision quality must be traded off (Li & Zhu, 2024; Masin & Bukchin, 2008).

## 2 PROBLEM SETUP

Let $X \in \mathcal{X}$ denote observed covariates, and let $Y \in \mathcal{Y}$ be a random outcome variable representing uncertain objective parameters. The decision-making task is formalized as solving a general constrained optimization problem, whose solution set is given by:

$$\pi(Y; \theta) := \arg\min_{z \in \mathcal{Z}(\theta)} g(z, Y, \theta) \subseteq \mathcal{Z}(\theta). \tag{1}$$

Here, $\theta$ denotes known parameters of the objective function $g$ and the feasible region $\mathcal{Z}(\theta)$, with the latter assumed to be a compact subset of $\mathcal{Z}$. Our goal is to develop a distribution-free, data-driven method for estimating the probability that a prescribed decision $z$ is suboptimal. Specifically, given a candidate decision $z$, we aim to develop a risk measure $\alpha(z)$ that satisfies:

$$\mathbb{P}\{z \in \pi(Y;\theta)\} \geq 1 - \alpha(z), \quad \forall z \in \mathcal{Z}, \tag{2}$$

where we omit the dependency of $\alpha(z)$ on $x$ for notation simplicity. The randomness on the left-hand side is induced by the $\pi(Y;\theta)$, which is dependent on the random variable $Y$. The objective in equation 2 is well-defined for any optimization problem and candidate decision satisfying $\mathbb{P}\{z \in \pi(Y;\theta)\} > 0$. Otherwise, one can take $\hat{\alpha}(z) = 1$ as the trivial solution[1].

## 3  CREDO: CONFORMALIZED DECISION RISK ASSESSMENT

We propose a framework for quantifying decision risk $\alpha(z)$ by casting this problem as a structured uncertainty quantification task over the decision space, as illustrated by Figure 2. The key idea is to project the decision's optimality condition to an *inverse feasible region* (Chan et al., 2025; Tavaslıoğlu et al., 2018) in the outcome space, and then construct inner geometric approximations of this region using *conformal calibrated* regions (Shafer & Vovk, 2008; Wang et al., 2023), where their miscoverage rates will be finally used as the risk estimates. We detail this procedure in two steps.

**Step 1: Reformulation with Inverse Feasible Region**    For any given realization $y$ of $Y$, a decision $z$ is optimal if and only if it achieves the smallest objective value among all feasible decisions:

$$z \in \pi(y;\theta) \quad \Longleftrightarrow \quad g(z,y;\theta) \leq g(z',y;\theta), \quad \forall z' \in \mathcal{Z}(\theta).$$

Therefore, the *inverse feasible region*, which is defined as the set of outcomes $y$ for which $z$ is an optimal decision, can be defined as

$$\pi^{-1}(z;\theta) := \bigcap_{z' \in \mathcal{Z}(\theta)} \{y \in \mathcal{Y} \mid g(z,y;\theta) \leq g(z',y;\theta)\}. \tag{3}$$

The definition equation 3 allows the objective to be reformulated as summarized in Proposition 1.

**Proposition 1** (Reformulation). *Let $\pi^{-1}(z;\theta)$ be defined in equation 3, then the objective defined in equation 2 can be equivalently expressed as:*

$$\mathbb{P}\{z \in \pi(Y;\theta)\} \equiv \mathbb{P}\{Y \in \pi^{-1}(z;\theta)\}. \tag{4}$$

The reformulated objective in equation 4 separates the random variable $Y$ from the mapping $\pi$. As a result, the original problem reduces to a standardized uncertainty quantification task, where the goal is to estimate the probability that $Y$ lies within $\pi^{-1}(z;\theta)$.

**Step 2: Risk Estimation via Generative Conformal Prediction**    To estimate the reformulated objective in equation 4, we propose to do so by constructing calibrated inner approximation sets of $\pi^{-1}(z;\theta)$. Specifically, we require two core conditions to be satisfied by the constructed set: ($a$) It is fully contained within $\pi^{-1}(z;\theta)$; ($b$) Its coverage probability of $Y$ is known or can be quantified. This allows us to use this constructed set as a surrogate to bound the reformulated objective by:

$$\mathbb{P}\{Y \in \pi^{-1}(z;\theta)\} \overset{(a)}{\geq} \mathbb{P}\{Y \in \mathcal{C}(X;\alpha)\} \overset{(b)}{\geq} 1 - \alpha. \tag{5}$$

Here, we denote $\mathcal{C}(X;\alpha)$ as the constructed set given input $X$, and $\alpha$ is the coverage probability, which should also serve as an estimate for $\alpha(z)$ in the objective defined in equation 2.

Additionally, the estimate $\alpha$ should not be overly conservative. This requires constructing approximation sets that not only fully exploit the space within $\pi^{-1}(z;\theta)$ but also concentrate around the high-density regions of $Y$, ensuring that both inequalities ($a$) and ($b$) in equation 5 are tight enough. A possible idea is to employ generative models as base predictors to capture the underlying distribution,

---

[1]Another option is to generalize the objective so that the near-optimal candidate decisions are also considered as a solution. This can be achieved by relaxing the optimal objective value by a prespecified small margin, see Andrews & Chen (2025); Kiyani et al. (2025).

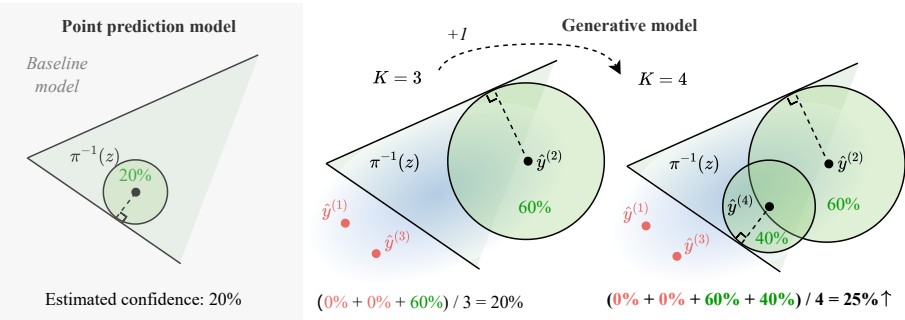

Figure 3: Visualization of decision risk estimation under different modeling choices. The figure compares point prediction model with generative models ($K = 3$ and $K = 4$) as the base prediction model, respectively. The black lines denote the boundary of the inverse feasible region $\pi^{-1}(z)$. The blue shade represents the true conditional distribution $Y \mid X$, and the black dots indicate generated predictions $\hat{y}^{(k)}$. The green ball indicates the conformalized sets $\mathcal{C}^{(k)}(x; \hat{\alpha}(z))$.

and then generate multiple and diverse conformalized sets as the approximation sets, which should empirically satisfy both conditions when the generative models are well-trained. We refer to this as the *generative conformal prediction* procedure, and it is formally detailed as follows.

We begin with training a (conditional) generative model $\hat{f} : \mathcal{X} \to \mathcal{Y}$ on a training dataset to approximate the conditional distribution of $Y|X$ and allow samples to be drawn from it. For a test input $x_{n+1}$, we draw a prediction $\hat{y}_{n+1} \sim \hat{f}(x_{n+1})$ and construct the *conformalized set* as an $\ell_2$ ball centered at $\hat{y}_{n+1}$:

$$\mathcal{C}(x_{n+1}; \alpha) = \left\{ y \in \mathcal{Y} \mid \|y - \hat{y}_{n+1}\|_2 < \hat{R}(\alpha) \right\} \text{ if } \alpha < 1 \text{ else } \emptyset, \tag{6}$$

where $\hat{R}(\alpha)$ denotes the *conformalized radius*, which is calibrated using a dataset $\{(x_i, y_i)\}_{i=1}^n$ to guarantee that $\mathcal{C}(x_{n+1}; \alpha)$ achieves valid $(1 - \alpha)$ coverage of the distribution of $Y$. For instance, one possible configuration of $\hat{R}(\alpha)$ is:

$$\hat{R}(\alpha) = \frac{\sum_{i=1}^n \|\hat{y}_i - y_i\|_2}{\alpha(n + 1) - 1}. \tag{7}$$

The coverage level $\alpha$ is determined by solving the following optimization problem, which finds the smallest $\alpha$ such that the *conformalized set* is entirely contained within the inverse feasible region:

$$\hat{\alpha}(z) = \min_{\alpha \in [1/(n+1), 1]} \left\{ \alpha \mid \mathcal{C}(x_{n+1}; \alpha) \subseteq \pi^{-1}(z; \theta) \right\}. \tag{8}$$

The procedure is repeated $K$ times to obtain the collection of estimates $\{\hat{\alpha}^{(k)}(z)\}_{k=1}^K$, which are then averaged to yield the final estimator:

$$\hat{\alpha}(z) = \left( \hat{\alpha}^{(1)}(z) + \ldots + \hat{\alpha}^{(K)}(z) \right) / K. \tag{9}$$

The full pseudocode of the algorithm is provided in Appendix A.

**Remark 1.** *We further clarify the role of generative models by emphasizing their advantages over point prediction models (*e.g.*, regressors). As shown in Figure 3, point prediction models can produce outputs that lie near the boundary of, or even outside, $\pi^{-1}(z; \theta)$, leading to overly conservative risk estimates (*i.e.*, risk equal to one). In contrast, generative models allow multiple draws, and increasing $K$ raises the chance that at least one prediction falls within the inverse feasible region, resulting in a more accurate (less than one) risk estimate. This idea will later be formalized through the notion of the true positive rate in our theoretical analysis.*

## 4  THEORETICAL ANALYSIS

This section presents three key theoretical analyses of our method. First, we prove conservativeness, showing that the estimator provides a valid upper bound of the decision risk. Second, we provide a

unifying view on the proposed risk estimator with Monte Carlo estimators. Finally, we show that the method supports high-quality decision-making, with its true positive rate increasing as the number of generated predictions grows. In our analyses, we set the conformalized radius as equation 7.

Our first theoretical result on conservativeness relies on the assumption that the calibration data are *exchangeable*. This requirement is milder than the i.i.d. assumption, as the latter implies the former, and is therefore commonly adopted in the conformal prediction literature (Angelopoulos & Bates, 2021; Barber et al., 2023; Papadopoulos et al., 2002).

**Assumption 1** (Exchangeability). *A dataset* $\{(x_i, y_i)\}_{i=1}^{n+1}$ *is exchangeable if, for any permutation* $\rho$ *of the index set* $\{1, \ldots, n+1\}$ *and all measurable sets* $\mathcal{A} \subseteq (\mathcal{X} \times \mathcal{Y})^{n+1}$*, there is*

$$\mathbb{P}\left\{(x_1, y_1), \ldots, (x_{n+1}, y_{n+1}) \in \mathcal{A}\right\} = \mathbb{P}\left\{(x_{\rho(1)}, y_{\rho(1)}), \ldots, (x_{\rho(n+1)}, y_{\rho(n+1)}) \in \mathcal{A}\right\}.$$

Under this assumption, we can derive Theorem 1, which states that our estimator yields a valid upper bound on the decision risk in expectation. Notably, this result does not require any assumption on how well the generative model captures the underlying distribution, which underscores the robustness of our algorithm against potential model misspecification, as well as the trustworthiness of the algorithm for applications where conservativeness is critical.

**Theorem 1** (Conservatism). *Under Assumption 1 the estimator* $\hat{\alpha}(z)$ *defined in equation 9 satisfies:*

$$\mathbb{P}\left\{z \in \pi(Y_{n+1}; \theta)\right\} \geq 1 - \mathbb{E}\left[\hat{\alpha}(z)\right].$$

Theorem 1 shows that the proposed estimator $\hat{\alpha}$ is, in expectation, a conservative estimate of the decision risk. The success of the proof relies on the post-hoc validity property of e-value conformal prediction (Vovk, 2025; Balinsky & Balinsky, 2024; Gauthier et al., 2025b), which motivates our design of the conformalized radius in equation 7. The detailed proof, along with a discussion of the first-order Taylor approximation procedure used therein, is provided in Appendix B.

Next, we highlight an important theoretical insight: the proposed estimator can be interpreted as a weighted Monte Carlo (MC) probability estimator based on $K$ generated predictions.

**Proposition 2.** *The estimator in equation 9 can be re-written as:*

$$\hat{\alpha}(z) = 1 - \frac{1}{K} \sum_{k=1}^{K} \left( w^{(k)}(z, x_{n+1}) \cdot \mathbb{1}\left\{\hat{y}_{n+1}^{(k)} \in \pi^{-1}(z; \theta)\right\}\right). \tag{10}$$

Here, the conformalized weights $w^{(k)}(z, x_{n+1}) \in [0, 1]$ are determined by our conformalization procedure, ensuring the conservatism guarantee established in Theorem 1. The unifying view in Proposition 2 hints potential extensions: by setting the conformalized weights to 1, we get a more radical variant of the estimator with more accurate risk estimation[2], but may fail the conservatism guarantee. We will examine this ablation variant in our subsequent synthetic experiments. The proof of Proposition 2 is provided in Appendix C.

Finally, we study the estimate's true positive rate (TPR), defined as the ratio of total decisions that are correctly estimated to have risks smaller than one out of all decisions with risks smaller than one:

$$\text{TPR} := \frac{\mathbb{E}\left[\#\{z \in \mathcal{Z}(\theta) \mid \alpha(z) < 1 \text{ and } \hat{\alpha}(z) < 1\}\right]}{\#\{z \in \mathcal{Z}(\theta) \mid \alpha(z) < 1\}}, \tag{11}$$

where with some abuse of notation, we denote $\alpha(z) = \mathbb{P}\{z \notin \pi(Y_{n+1}; \theta)\}$ as the actual risk. TPR indirectly reflects the discrepancy in final decisions led by different outputted risk estimates: assuming that decision-makers choose to rule out decisions $z$ with $\hat{\alpha}(z) = 1$, then for those "false-positive" rule-outs, the decision-maker would risk losing profits as they've omitted decisions that are possible to be optimal. Therefore, TPR scales positively with the quality of the decision assessment.

**Proposition 3** (True positive rate). *TPR (equation 11) monotonically increases as* $K$ *increases.*

Proposition 3 specifically highlights the generative model's important role in avoiding overconservatism, confirming the intuition built by Figure 3: when $K$ is small (*e.g.*, $K = 1$), or when the generative model is replaced by a deterministic predictor, the estimator becomes more susceptible to such erroneous exclusions, potentially leading to a lower TPR. Therefore, the generative approach of drawing diverse predictions would help mitigate this issue and support high-quality decision-making. These insights are also tested in our numerical experiments. Its proof is detailed in Appendix D.

---

[2]Assuming $K$ is large and the predictive model $\hat{f}(Y|X)$ is well-trained.

## 5 EFFICIENT COMPUTATION FOR LINEAR PROGRAMS

Recall that the generic CREDO algorithm described in Section 3 makes no assumptions about the form of the objective function $g(z, Y; \theta)$ or the feasible region $\mathcal{Z}(\theta)$, enabling its application to general constrained optimization problems. In particular, under a linear programming setting, we can further show that the algorithm is highly computationally efficient as it admits a closed-form solution, thereby eliminating the need for iterative or approximation-based procedures.

Suppose $\mathcal{Y}, \mathcal{Z} \subseteq \mathbb{R}^d$, a linear programming (LP) problem is defined as:

$$\pi_{\text{LP}}(Y; \theta); =; \underset{z \in \mathcal{Z}(\theta)}{\arg\min} \langle Y, z \rangle, \quad \mathcal{Z}(\theta) = \{z \in \mathcal{Z} \mid \mathbf{A}z \leq b\}, \quad (12)$$

where $\mathbf{A} \in \mathbb{R}^{d \times m}$ and $b \in \mathbb{R}^m$ are known parameters belonging to $\theta$. As one of the most fundamental optimization structures, LPs arise naturally in a wide range of real-world applications, including supply chain management, energy dispatch, and transportation (Charnes & Cooper, 1957). Moreover, LPs are often employed to approximate more complex optimization problems, with many relaxations proving both useful and widely applied in practice (Lovász, 1975). The closed-form expression of our estimator in this setting follows directly as a corollary to Equation (10):

**Corollary 1.** *Suppose the optimization problem is defined in equation 12, then:*

$$\hat{\alpha}(z) = 1 - \frac{1}{K} \sum_{k=1}^{K} \left[ \left( \frac{nD^{(k)} - \sum_{i=1}^{n} \|\hat{y}_i - y_i\|_2}{(n+1)D^{(k)}} \right)^+ \prod_{v \in \mathcal{V}(\theta)} \mathbb{1}\left\{ \langle \hat{y}_{n+1}^{(k)}, z - v \rangle \leq 0 \right\} \right], \quad (13)$$

*where $\mathcal{V}(\theta)$ denotes the set of vertices of $\mathcal{Z}(\theta)$, and $\hat{D}^{(k)}$ is the distance of $\hat{y}_{n+1}^{(k)}$ to the boundary of $\pi^{-1}(z; \theta)$, with the expression $\hat{D}^{(k)} = \min_{v \in \mathcal{V}(\theta) \setminus \{z\}} |\langle \hat{y}_{n+1}^{(k)}, z - v \rangle| / \|z - v\|_2$.*

In Corollary 1, computing vertices of a polytope $\mathcal{Z}(\theta)$ can be efficiently executed with well-established algorithms such as the double description method (Fukuda & Prodon, 1995; Motzkin et al., 1953). Once the vertices are precomputed, the estimator $\hat{\alpha}(z)$ can be computed via a single pass over equation 13, entirely avoiding set enumerations and numerical optimizations. This renders it highly efficient for large-scale LP problems, with a computational complexity of $\mathcal{O}(K \cdot n \cdot |\mathcal{V}(\theta)|)$ that is independent of the number of iterations/epochs. Moreover, one can prove that when $z \notin \mathcal{V}(\theta)$, the indicator term turns out to be zero (Tavaslıoğlu et al., 2018). As a result, the algorithm can be further accelerated by automatically outputting risk one if it detects that the input decision is not on the vertex of the feasible region, making our proposed algorithm computationally efficient and implementation-friendly. The proof of Corollary 1 is provided in Appendix C.

We emphasize that although closed-form characterizations are available only for LPs, the algorithm described in Section 3 is more broadly computationally feasible to general convex optimization problems. In particular, following Proposition 2, one can construct the risk estimator by: $(i)$ solving a forward optimization problem to determine whether $\hat{y}_{n+1}^{(k)}$ lies in the inverse feasible region associated with decision $z$, and $(ii)$ computing the conformalized weight $w^{(k)}(z, x_{n+1})$ as defined in Appendix C.

## 6 EXPERIMENTS

In this section, we evaluate CREDO on both synthetic and real-world optimization problems. We show that: $(i)$ The proposed risk estimator is empirically conservative, adding to Theorem 1; $(ii)$ The generative approach improves the accuracy of CREDO in risk estimates, adding to Proposition 3; $(iii)$ How the key parameters, such as the number of generated samples $K$ and variance scales $\sigma$, impact the performance of CREDO; $(iv)$ CREDO produces high quality decisions under various settings.

Our experiment settings include: $(i)$ Two synthetic settings that are referred to as Setting I and Setting II, illustrated in Figure 4. Setting I features a triangular feasible region with three vertices and can be interpreted as a basic profit-maximization problem. In contrast, Setting II presents a more stylized scenario with an octagonal feasible region comprising five vertices. $(ii)$ A real-world infrastructure planning problem, where we study a budget-constrained substation upgrade problem, formulated as a knapsack optimization, that aims to minimize the expected overflow of solar panel installations

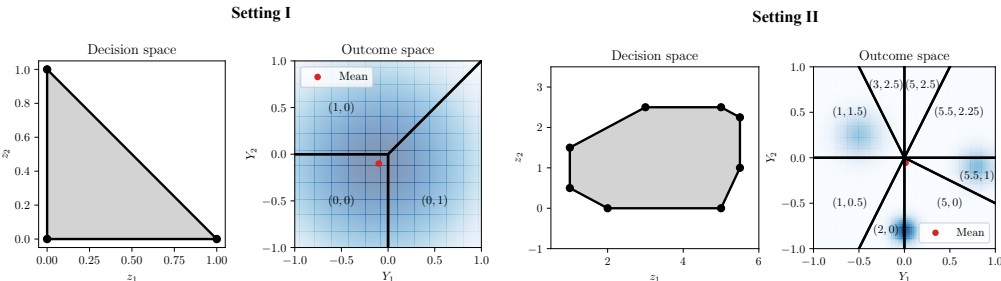

Figure 4: Illustration of settings I and II. The gray region represents the feasible region in the decision space, and the cones to the right are the corresponding inverse feasible regions in the outcome space. The blue shade denotes the density mass of $Y$.

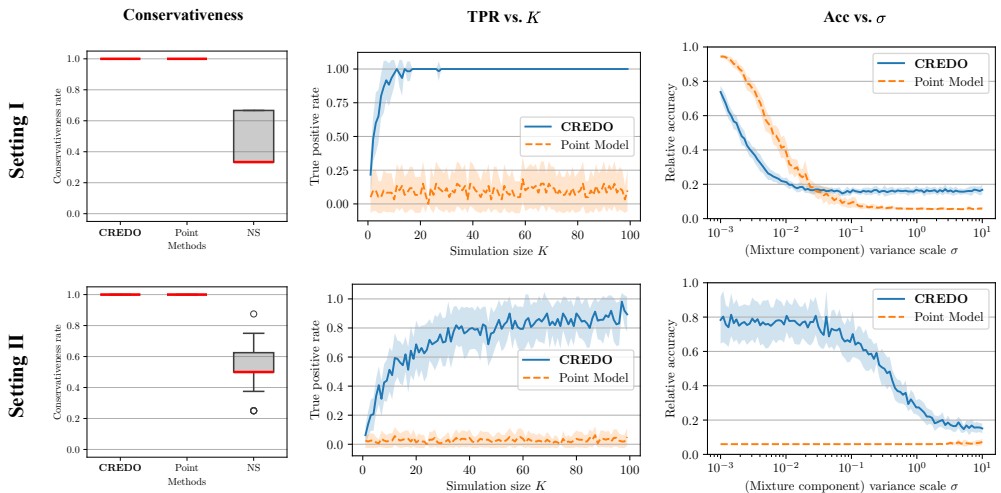

Figure 5: Selected results from the ablation comparison. Each column shows (from left to right): Conservativeness of different ablation models; True positive rate (TPR) versus generative sample size $K$; Relative accuracy versus variance scale $\sigma$. Experiments are repeated across 20 independent trials. The default configuration is $K = 100$ and $\sigma = 1$ unless otherwise specified.

in a power distribution grid (Zhou et al., 2024). We use real data of solar panel installation records collected 2010-2024 from a utility situated in Indiana, U.S. More details can be found in Appendix E.

In our experiments, we specify the generative model as a three-component Gaussian Mixture model, fitted using the EM algorithm for $1 \times 10^2$ epochs. This modeling choice balances its capability of accurately capturing multi-modality in the data, without the need for a large amount of training data as required by deep learning architectures. The training and calibration data are randomly and equally split following the split conformal prediction framework (Papadopoulos et al., 2002). Across all experiments, we adopt the standard p-value conformalized radius (Singh et al., 2024) as the conformalized radius $\hat{R}(\alpha)$, which has been demonstrated by prior works to show strong empirical accuracy and conservativeness properties. More details can be found in Appendix E.

**Conservativeness vs. Accuracy Tradeoff** In the first set of experiments, we evaluate the properties of CREDO through controlled component analysis under the synthetic settings. We focus on testing for two hypotheses: it can consistently produce conservative risk estimates, and it achieves superior accuracy, meaning that it can closely estimate the ground-truth risk.

We adopt three metrics[3]: ($i$) Conservativeness rate: the percentage of trials where the risk estimate satisfies the conservativeness guarantee in equation 2 for all feasible decisions; ($ii$) True positive rate (TPR): the percentage of actions where the method correctly identifies them to have risks of less than one, averaged across trials (equation 11); ($iii$) Relative accuracy (Acc): the negative normalized sum

---

[3]Note that the latter two both aim at assessing the accuracy.

of absolute differences between the estimated and true risks across all decisions, averaged across trials.

We compare the performance of CREDO on these metrics with two direct ablation variants: (*i*) Point: a variant that uses a point prediction (the conditional mean) instead of sampling from a full probabilistic model (Singh et al., 2024). (*ii*) NS: the naive sampling estimator that sets the conformalized weights to one in equation 10, taking a Monte Carlo probability estimator form.

We present some selected insightful results in Figure 5 and leave the complete results to Appendix F. It can be observed from the first column that both CREDO and Point consistently achieve 100% conservative estimates, whereas NS attains only around 50%. These results align with Theorem 1, highlighting the significance of the conformalized weighting term and its effectiveness in downscaling the estimate to satisfy the conservativeness guarantee. Due to the deficiency of NS in conservativeness, which is the key property that we study for, it has been withdrawn from subsequent comparisons.

In the second column of Figure 5, we observe that as the sample size $K$ increases, CREDO exhibits a significant increase in true positive rate. In contrast, the curve for Point remains flat. This indicates that, while maintaining a similar level of conservativeness, CREDO can identify a greater number of potentially suboptimal actions, whereas Point may overlook viable alternatives that the decision-maker could consider. This serves as strong numerical evidence supporting Proposition 3.

In the third column of Figure 5, we observe that in Setting I, as the data variance scale increases, the performance of CREDO crosses over and eventually surpasses that of Point. In Setting II, CREDO consistently achieves higher relative accuracy across all variance levels. These results suggest that CREDO is particularly effective when the outcome variable $Y$ is highly stochastic and challenging to approximate with point estimators. This further validates the underlying motivation of incorporating generative components instead of point-prediction models.

**Decision Quality Evaluation**    In the second experiment, we assess decision quality based on risk estimates produced by CREDO across both synthetic and real-world settings. We show that CREDO can be used to select decisions with consistently higher confidence, demonstrating the effectiveness of CREDO's risk estimates in guiding practical decision-making.

We adopt *empirical confidence ranking* as our primary evaluation metric: given a decision policy $\pi$ and a test dataset $\{(x_i, y_i)\}_{i=1}^m$, we apply $\pi$ to each input $x_i$ to generate predicted decisions $\{z_i\}_{i=1}^m$. We then compute the score $\sum_{i=1}^m h(z_i)$, where $h$ maps each prediction to a discrete rank based on its frequency among the ground-truth optimal decisions $\{z_i^*\}_{i=1}^m$ in the test set. This metric is designed to capture a method's tendency to select decisions that are most likely to be optimal. It is instance-independent and would discourage decisions that are rare in the optimal set of actions in the test set, which is consistent with what a decision-maker would desire for a prescribed decision to achieve when operating under ex-ante uncertainty. Furthermore, the metric is unit-independent and is robust to extreme outliers, which renders it more appropriate for a one-shot decision-making scheme considered in our paper, compared to the standard notions such as regret.

We define the decision policy for CREDO as selecting the action with the lowest estimated risk. It is compared with four decision-making baselines: predict-then-optimize (PTO) (Bertsimas & Kallus, 2020), robust optimization (RO) (Bertsimas & Thiele, 2006), smart PTO (SPO+) (Elmachtoub & Grigas, 2022), and decision-focused learning (DFL) (Amos & Kolter, 2017). These baselines are chosen for their popularity and their alignment with the prescriptive, risk-averse decision-making paradigm underlying our decision risk assessment setup.

Table 1: Evaluated empirical confidence ranking ($\downarrow$) for different methods across three datasets.

| Method | Setting I | | | Setting II | | | Real Data |
|---|---|---|---|---|---|---|---|
| | $\sigma = 0.1$ | $\sigma = 1$ | $\sigma = 10$ | $\sigma = 0.1$ | $\sigma = 1$ | $\sigma = 10$ | |
| PTO | $\mathbf{1.00 \pm 0.00}$ | $2.76 \pm 0.59$ | $2.24 \pm 0.79$ | $3.55 \pm 0.50$ | $3.36 \pm 0.48$ | $2.04 \pm 1.65$ | $1.75 \pm 1.69$ |
| RO | $\mathbf{1.00 \pm 0.00}$ | $2.98 \pm 0.14$ | $3.00 \pm 0.00$ | $4.99 \pm 0.10$ | $6.00 \pm 0.00$ | $3.98 \pm 0.80$ | $3.00 \pm 1.29$ |
| SPO+ | $\mathbf{1.00 \pm 0.00}$ | $2.68 \pm 0.65$ | $2.02 \pm 0.82$ | $3.95 \pm 1.20$ | $4.67 \pm 1.56$ | $3.56 \pm 1.50$ | $2.67 \pm 1.43$ |
| DFL | $2.44 \pm 0.64$ | $1.83 \pm 0.81$ | $2.06 \pm 0.79$ | $3.60 \pm 1.52$ | $3.96 \pm 2.07$ | $3.66 \pm 2.48$ | $1.92 \pm 1.04$ |
| CREDO | $1.75 \pm 0.77$ | $\mathbf{1.61 \pm 0.56}$ | $\mathbf{1.48 \pm 0.52}$ | $\mathbf{1.05 \pm 0.22}$ | $\mathbf{1.00 \pm 0.00}$ | $2.03 \pm 0.96$ | $1.75 \pm 0.92$ |

Table 1 presents the comparison results. It can be seen that CREDO achieves the smallest ranking metric value across most datasets, on average selecting the top two most likely decisions across all datasets. Though it might seem concerning that the in Setting I ($\sigma = 0.1$), PTO, RO, and SPO+ all achieve better performance than CREDO, this is because when $\sigma$ is small, the data becomes highly concentrated around the mean, rendering the problem nearly deterministic and can be best dealt with point-prediction baselines. This also mirrors the behavior observed in the Accuracy vs. $\sigma$ plot for Setting I in Figure 5. These results highlight the capability of our method for high-quality decision-making, especially under highly uncertain environments.

## 7    CONCLUSION

We proposed CREDO, a distribution-free framework for decision risk assessment that combines inverse optimization with conformal prediction to estimate the probability that a candidate decision is suboptimal. Our method provides statistically valid risk certificates by characterizing inverse optimality regions and using generative models to construct calibrated inner approximations. Empirical results on synthetic and real-world datasets demonstrate that CREDO offers conservative yet informative risk estimates, enabling more robust and interpretable decision support under uncertainty.

From a practical standpoint, the conformalized radius in CREDO is not restricted to the e-value variant discussed in this work. The e-value approach offers rigorous post-hoc validity guarantees, but alternatives such as the p-value variant provide tighter risk estimates at the cost of weakened validity. Selecting an appropriate radius thus requires balancing the trade-off between validity and informativeness, ensuring that decision support remains both safe and actionable.

We also highlight a potential concern regarding *selection bias*. When the risk estimate produced by CREDO is used to select a decision, re-evaluating the risk of that selected decision may invalidate the original guarantee. This arises because our current framework implicitly assumes that conditioning on the input decision doesn't violate the exchangeability of the calibration data. Once this assumption is broken, the theoretical validity of the risk estimate can no longer be ensured. A simple mitigation strategy is to use data splitting, separating the data used for decision selection from that used for risk evaluation. Additionally, we encourage future work to address this selection bias without relying on sample splitting, potentially by developing principled relaxations or adjustments of the risk guarantees that explicitly account for data-dependent decision selection.

**Ethics Statement**    This work does not involve human subjects, sensitive personal data, or applications that pose direct risks of harm. All datasets used in this paper, along with an anonymized codebase, are provided in the supplementary material, with any potentially sensitive information removed from the real data. We ensured that our methods were applied in a way that does not introduce or amplify unfair bias, discrimination, or privacy risks. The research was conducted in accordance with the ICLR Code of Ethics.

**Reproducibility Statement**    We have taken steps to ensure the reproducibility of our results. Experimental details, including dataset descriptions, preprocessing, model architectures, hyperparameters, and evaluation metrics, are documented in the main text and the appendix. We have included clear explanations of any assumptions and complete proofs of the theoretical results in both the main paper and the appendix. An anonymized codebase with implementation and notebooks for reproducing our experiments is included in the supplementary material.

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

APPENDIX OVERVIEW

In the Appendix, we provide comprehensive supporting material for the main manuscript. We include a detailed description of the algorithm pseudo code (Appendix A), detailed proof and derivation for all theoretical results (Appendices B to D), detailed experiment setups Appendix E, and additional experiment results Appendix F.

**Remark 2.** *Throughout the theoretical derivations, we introduce slightly modified versions of some definitions of the algorithmic components in Section 3. In particular, we redefine the conformalized set (equation 6), conformalized radius (equation 7), and estimated risk (equation 8) respectivly as*

$$\mathcal{C}(x_{n+1}; \alpha) = \left\{ y \in \mathcal{Y} \,\middle|\, \|y - \hat{y}_{n+1}\|_2 < \hat{R}(\alpha) \right\},$$

$$\hat{R}(\alpha) = \begin{cases} +\infty & \text{if } \alpha \in [0, \frac{1}{n+1}), \\ \frac{\sum_{i=1}^{n} \|\hat{y}_i - y_i\|_2}{\alpha(n+1) - 1} & \text{if } \alpha \in [\frac{1}{n+1}, 1), \\ 0 & \text{if } \alpha = 1. \end{cases}$$

$$\hat{\alpha}(z) = \min_{\alpha \in [0,1]} \left\{ \alpha \,\middle|\, \mathcal{C}(x_{n+1}; \alpha) \subseteq \pi^{-1}(z; \theta) \right\}.$$

*Compared to the original definitions, these modified versions are more convenient for mathematical derivations, though slightly less interpretable. It is straightforward to verify that adopting these modifications does not affect the* CREDO *algorithm or its associated theoretical guarantees.*

## A  ADDITIONAL ALGORITHM DETAILS

The algorithm pseudo-code of the conformalized decision risk assessment described in the main text is summarized in Algorithm 1.

Additionally, we propose a heuristic algorithm for solving $\hat{\alpha}^{(k)}(z)$ in Step 9, with the algorithm pseudo-code summarized in Algorithm 2. Specifically, we note that the user needs to specify two lists of finite points $\tilde{\mathcal{Y}} \subseteq \mathcal{Y}$ and $\tilde{\mathcal{Z}} \subseteq \mathcal{Z}(\theta)$ a priori, which serve as approximations for their original space counterparts. Some examples that can be considered for $\tilde{\mathcal{Y}}$ include:

- When $|\mathcal{Y}| < +\infty$, one can trivially take $\tilde{\mathcal{Y}} = \mathcal{Y}$.
- When $\mathcal{Y}$ is a bounded metric space, one can take $\tilde{\mathcal{Y}}$ as the $\epsilon$-net of $\mathcal{Y}$ (*a.k.a.* the vertices of some finite grid discretization).
- When $\mathcal{Y}$ is an unbounded metric space, one can set $\tilde{\mathcal{Y}}$ as the $\epsilon$-net of the level set of the marginal distribution of $Y$. Namely, for some $\beta \geq 0$, this level set is denoted as: $\{y \in \mathcal{Y} \mid p_Y(y) > \beta\}$. It indicates the regions where $Y$ has a density mass of at least $\beta \geq 0$.

For $\mathcal{Z}(\theta)$, since we have already assumed it to be a compact space, then the first two options can be similarly used when constructing $\tilde{\mathcal{Z}}$.

## B  PROOF OF THEOREM 1

Conditioned on the modified definitions introduced in Remark 2, we first present the following lemma:

**Lemma 1** (E-value post-hoc validity)**.** *Under Assumption 1, then the estimator $\hat{\alpha}(z)$ satisfies*

$$\mathbb{P}\left\{ Y_{n+1} \in \mathcal{C}^{(k)}(X_{n+1}; \hat{\alpha}) \right\} \geq 1 - \mathbb{E}[\hat{\alpha}], \quad \forall k = 1, \ldots, K.$$

*where $\hat{\alpha}$ can probabilistically depend on $\{(X_i, Y_i)\}_{i=1}^{n+1}$ in arbitrary ways.*

*Proof of Lemma 1.* When $\hat{\alpha} < 1/(n+1)$ or $\hat{\alpha} = 1$, by the definition of $\hat{R}(\alpha)$, the statement holds trivially. So we only need to consider the case when $1 > \hat{\alpha} \geq 1/(n+1)$. For any $k = 1, \ldots, K$, we

---

**Algorithm 1** Conformalized Decision Risk Assessment (`CREDO`)

---

**Require:** Fitted generative model $\hat{f}$; Calibration dataset $\{(x_i, y_i)\}_{i=1}^n$; Sample size $K$; Optimization parameter $\theta$; Decision $z$; Test covariate $x_{n+1}$.
1: $\pi^{-1}(z; \theta) \leftarrow$ Compute the inverse feasible region defined in equation 3;
2: Initialize nonconformity score set $\mathcal{E} \leftarrow \emptyset$;
3: **for** $i \in \{1, \ldots, n\}$ **do**
4:    $\hat{y}_i \sim \hat{f}(x_i)$; $r_i \leftarrow \|y_i - \hat{y}_i\|_2$; $\mathcal{E} \leftarrow \mathcal{E} \cup \{r_i\}$;
5: **end for**
6: **for** $k = 1, \ldots, K$ **do**
7:    $\hat{y}_{n+1}^{(k)} \sim \hat{f}(x_{n+1})$;
8:    $\mathcal{C}^{(k)}(x_{n+1}; \alpha) \leftarrow$ Construct conformal set given $\mathcal{E}$ and $\hat{y}_{n+1}^{(k)}$ via equation 6;
9:    $\hat{\alpha}^{(k)}(z) \leftarrow$ Solve for the $k$-th decision risk via equation 8 (refer to Algorithm 2);
10: **end for**
11: **return** $\hat{\alpha}(z) \leftarrow 1/K \cdot \sum_{k=1}^K \hat{\alpha}^{(k)}(z)$.

---

**Algorithm 2** Heuristic algorithm for risk estimation

---

**Require:** Proposal points $\tilde{\mathcal{Y}}$ and $\tilde{\mathcal{Z}}$; Required inputs of Algorithm 1.
1: $\tilde{\pi}_c^{-1}(z; \theta) \leftarrow \bigcap_{z' \in \tilde{\mathcal{Z}}} \left\{ y \in \tilde{\mathcal{Y}} \mid g(z, y; \theta) > g(z', y; \theta) \right\}$.
2: $\tilde{D}^{(k)} \leftarrow \min_{\tilde{y} \in \tilde{\pi}_c^{-1}(z; \theta)} \|\tilde{y} - \hat{y}_{n+1}^{(k)}\|_2$.
3: $\tilde{\alpha}^{(k)}(z) \leftarrow \frac{\tilde{D}^{(k)} + \sum_{i=1}^n \|\hat{y}_i - y_i\|_2}{(n+1)\tilde{D}^{(k)}}$.
4: **return** Approximated risk estimator $\tilde{\alpha}^{(k)}(z)$.

---

begin by expanding the left-hand side:

$$\mathbb{P}\left\{ Y \notin \mathcal{C}^{(k)}(X; \hat{\alpha}) \right\} = \mathbb{P}\left\{ \|\hat{y}_{n+1}^{(k)} - Y\|_2 > \frac{\sum_{i=1}^n \|\hat{y}_i - Y_i\|_2}{\hat{\alpha}(n+1) - 1} \right\}$$

$$= \mathbb{P}\left\{ \hat{\alpha}(n+1)\|Y - \hat{y}_{n+1}^{(k)}\|_2 > \sum_{i=1}^n \|\hat{y}_i - Y_i\|_2 + \|\hat{y}_{n+1}^{(k)} - Y\|_2 \right\}$$

$$= \mathbb{P}\left\{ \hat{\alpha} > \frac{\sum_{i=1}^n \|\hat{y}_i - Y_i\|_2 + \|\hat{y}_{n+1}^{(k)} - Y\|_2}{(n+1)\|\hat{y}_{n+1}^{(k)} - Y\|_2} \right\}$$

$$= \mathbb{P}\left\{ \frac{(n+1)\|\hat{y}_{n+1}^{(k)} - Y\|_2}{\sum_{i=1}^n \|\hat{y}_i - Y_i\|_2 + \|\hat{y}_{n+1}^{(k)} - Y\|_2} > \frac{1}{\hat{\alpha}} \right\}.$$

Denote the following random variables,

$$F_i = \frac{(n+1)\|\hat{y}_i - Y_i\|_2}{\sum_{i=1}^n \|\hat{y}_i - Y_i\|_2 + \|\hat{y}_{n+1}^{(k)} - Y\|_2}, \quad \forall i = 1, \ldots, n,$$

$$F_{n+1} = \frac{(n+1)\|\hat{y}_{n+1}^{(k)} - Y\|_2}{\sum_{i=1}^n \|\hat{y}_i - Y_i\|_2 + \|\hat{y}_{n+1}^{(k)} - Y\|_2}.$$

It can be seen that the following two conditions hold:

$$(a): \quad \mathbb{E}[F_1 + \ldots + F_n + F_{n+1}] = n + 1,$$
$$(b): \quad \mathbb{E}[F_1] = \ldots = \mathbb{E}[F_n] = \mathbb{E}[F_{n+1}],$$

where $(b)$ holds by exchangeability (Assumption 1). Therefore, there is

$$\mathbb{E}[F_{n+1}] = 1. \tag{14}$$

Using this result, it can be derived that

$$\sup_{\tilde{\alpha}} \mathbb{E}\left[ \frac{\mathbb{P}(F_{n+1} > 1/\tilde{\alpha})}{\tilde{\alpha}} \right] \leq \sup_{\tilde{\alpha}} \mathbb{E}\left[ \frac{\tilde{\alpha} \cdot \mathbb{E}[F_{n+1}]}{\tilde{\alpha}} \right] = \mathbb{E}[F_{n+1}] = 1,$$

where the inequality follows from Markov's inequality. Therefore, for any $\hat{\alpha}$ which may depend on the data $\{X_i, Y_i\}_{i=1}^{n+1}$, there is:

$$\mathbb{E}\left[\frac{\mathbb{P}(F_{n+1} > 1/\hat{\alpha} \mid \hat{\alpha})}{\hat{\alpha}}\right] \leq \sup_{\tilde{\alpha}} \mathbb{E}\left[\frac{\mathbb{P}(F_{n+1} > 1/\tilde{\alpha})}{\tilde{\alpha}}\right] \leq 1$$

Using a first-order Taylor expansion on the left-hand side of the inequality above, there is

$$\mathbb{E}\left[\frac{\mathbb{P}(F_{n+1} > 1/\hat{\alpha} \mid \hat{\alpha})}{\hat{\alpha}}\right] \approx \frac{\mathbb{E}\left[\mathbb{P}(F_{n+1} > 1/\hat{\alpha} \mid \hat{\alpha})\right]}{\mathbb{E}[\hat{\alpha}]} \tag{15}$$

We assume that this approximation is exact, so the $\approx$ sign can be replaced with a $=$ sign (detailed discussion in Appendix B.1). Consequently, by combining the two equations above, we get

$$\mathbb{E}\left[\mathbb{P}(F_{n+1} > 1/\hat{\alpha} \mid \hat{\alpha})\right] \leq \mathbb{E}[\hat{\alpha}] \iff \mathbb{P}\left\{Y_{n+1} \in \mathcal{C}^{(k)}(X_{n+1}; \hat{\alpha})\right\} \geq 1 - \mathbb{E}[\hat{\alpha}],$$

where $\hat{\alpha}$ can probabilistically depend on $\{(X_i, Y_i)\}_{i=1}^{n+1}$ in arbitrary ways. This finishes the proof. $\square$

*Proof of Theorem 1.* We begin by observing that

$$\mathbb{P}\{z \in \pi(Y; \theta)\} = \mathbb{P}\left\{Y \in \pi^{-1}(z; \theta)\right\} \geq \frac{1}{K}\sum_{k=1}^{K}\mathbb{P}\left\{Y \in \mathcal{C}^{(k)}\left(X; \hat{\alpha}^{(k)}(z)\right)\right\}. \tag{16}$$

The first equality is due to the problem reformulation equation 4. The second inequality holds due to the definition of $\hat{\alpha}^{(k)}(z)$, which guarantees that the $k$-th generated conformal prediction region is always contained in $\pi^{-1}(z)$. Since by Lemma 1, we have proved that:

$$\mathbb{P}\left\{Y \in \mathcal{C}^{(k)}\left(X; \hat{\alpha}^{(k)}(z)\right)\right\} \geq 1 - \mathbb{E}\left[\hat{\alpha}^{(k)}(z)\right].$$

Therefore, combining this with equation 16 we obtain:

$$\mathbb{P}\{z \in \pi(Y; \theta)\} \geq 1 - \mathbb{E}\left[\frac{1}{K}\sum_{k=1}^{K}\hat{\alpha}^{(k)}(z)\right] = 1 - \mathbb{E}\left[\hat{\alpha}(z)\right].$$

We conclude the entire proof for Theorem 1. $\square$

### B.1 Discussion on the approximation error in equation 15

In this section, we comment on the approximation error of the first-order Taylor approximation in equation 15. This approximation trick has been adopted in prior works (Gauthier et al., 2025b), which has been argued that its error is small when the estimator $\hat{\alpha}$ is well concentrated around its mean. Empirically, this condition is usually satisfied in our setting. For example, when CREDO is deployed in a human-algorithm collaboration setting, the candidate decisions provided from the decision maker would be expected to be near optimal and should already enjoy a relatively small ground truth risk. This makes $\hat{\alpha}$ have a relatively small variance and well concentrated around its mean.

Even when this condition does not hold, we can resort to an alternative way to account for the approximation error during risk assessment. This can be done by theoretically deriving the approximation error and then manually offsetting the error in our risk estimator to achieve an exact conservativeness guarantee. Specifically, let $h(\hat{\alpha}) := \mathbb{E}[\mathbb{P}(F_{n+1} > 1/\hat{\alpha} \mid \hat{\alpha})]$, there is:

$$\left|\mathbb{E}\left[\frac{h(\hat{\alpha})}{\hat{\alpha}}\right] - \frac{\mathbb{E}[h(\hat{\alpha})]}{\mathbb{E}[\hat{\alpha}]}\right| \leq \mathbb{E}[h(\hat{\alpha})]\left(\mathbb{E}\left[\frac{1}{\hat{\alpha}}\right] - \frac{1}{\mathbb{E}[\hat{\alpha}]}\right) + \sqrt{\text{Var}(h(\hat{\alpha})) \cdot \text{Var}\left(\frac{1}{\hat{\alpha}}\right)}.$$

The first term is the Jensen gap, and the second term is by the Cauchy-Schwarz inequality. Assuming $\hat{\alpha} \in [\delta, 1]$ almost surely, then

$$\mathbb{E}\left[\frac{1}{\hat{\alpha}}\right] - \frac{1}{\mathbb{E}[\hat{\alpha}]} \leq \frac{1}{\delta^3}\text{Var}(\hat{\alpha}), \quad \text{and} \quad \text{Var}\left(\frac{1}{\hat{\alpha}}\right) \leq \frac{1}{\delta^3}\text{Var}(\hat{\alpha}).$$

Then, plugging them into the previous equation, we get

$$\left| \mathbb{E}\left[ \frac{h(\hat{\alpha})}{\hat{\alpha}} \right] - \frac{\mathbb{E}[h(\hat{\alpha})]}{\mathbb{E}[\hat{\alpha}]} \right| \leq \frac{1}{\delta^3} \operatorname{Var}(\hat{\alpha}) + \frac{1}{2\delta^2} \sqrt{\operatorname{Var}(\hat{\alpha})}.$$

Since for random variables bounded within $[\delta, 1]$, there is the following trivial upper bound:

$$\operatorname{Var}(\hat{\alpha}) \leq \frac{1}{K^2} \sum_{k,k'} \operatorname{Cov}(I_k, I'_k) \leq \frac{1}{4},$$

therefore, we can conclude that:

$$\left| \mathbb{E}\left[ \frac{\mathbb{P}(F_{n+1} > 1/\hat{\alpha} \mid \hat{\alpha})}{\hat{\alpha}} \right] - \frac{\mathbb{E}\left[ \mathbb{P}(F_{n+1} > 1/\hat{\alpha} \mid \hat{\alpha}) \right]}{\mathbb{E}[\hat{\alpha}]} \right| \leq \frac{1}{4\delta^3} + \frac{1}{4\delta^2}.$$

Plugging this result back into the proof of Theorem 1, we get

$$\mathbb{P}\left\{ z \in \pi(Y; \theta) \right\} \geq 1 - \mathbb{E}[\hat{\alpha}(z)] - 1/4(\delta^{-3} + \delta^{-2}).$$

Therefore, one can take the final estimator as

$$\min\{\hat{\alpha}(z) + 1/4(\delta^{-3} + \delta^{-2}), 1\} \tag{17}$$

so that exact conservativeness is achieved.

Note that in the derivation above, we have assumed that $\hat{\alpha} \in [\delta, 1]$ almost surely and $\delta$ is known. A trivial value that the user can take for $\delta$ is $1/(n+1)$, which is guaranteed by the design of the CREDO algorithm. We can also manually tune the value of $\delta$ by modifying the conformalized radius as

$$\hat{R}(\alpha) = \begin{cases} +\infty, & \text{if } \alpha \in [0, \delta), \\ \sum_{i=1}^n \|\hat{y}_i - y_i\|_2 / (\alpha(n+1) - 1) & \text{if } \alpha \in [\delta, 1), \\ 0 & \text{if } \alpha = 1, \end{cases}$$

to achieve a tighter bound (*i.e.*, smaller offset). One can prove that as long as $\delta$ is chosen such that $\delta > 1/(n+1)$, all theorems presented in the main text remain valid, and the bound (*i.e.*, offset) becomes tighter as $\delta$ increases. Then, one can take equation 17 as the final estimator.

In the meantime, we note that the above definition of the conformalized radius is equivalent to truncating the lower part of $\hat{\alpha}(z)$ at $\delta$, *i.e.*, setting $\max\{\hat{\alpha}(z), \delta\}$ as the risk estimator, and then taking equation 17 as the final estimator.

## C   PROOF OF PROPOSITION 2 AND COROLLARY 1

This section consists of three parts: we first prove the weighted Monte Carlo estimator form provided in equation 10 in Proposition 2; Then, we derive the indicator term; Finally, we derive the conformalized weighting term. The latter two all appear in Corollary 1.

Please note that all derivations use the modified definitions introduced in Remark 2.

*Proof of equation 10.* Note that by definition of the conformalized radius $\hat{R}(\alpha)$ and $\hat{\alpha}^{(k)}(z)$, when the $k$-th prediction falls outside of the inverse feasible region, then $\hat{\alpha}^{(k)}(z)$ would be conservatively set to one. Therefore, we can make the following decomposition:

$$\hat{\alpha}^{(k)}(z) = \mathbb{1}\left\{ \hat{y}_{n+1}^{(k)} \in \pi^{-1}(z; \theta) \right\} \cdot \min_{\alpha \in [0,1)} \left\{ \alpha \,\Big|\, \mathcal{C}^{(k)}(x_{n+1}; \alpha) \subseteq \pi^{-1}(z; \theta) \right\} + \mathbb{1}\left\{ \hat{y}_{n+1}^{(k)} \notin \pi^{-1}(z; \theta) \right\}$$

$$= \mathbb{1}\left\{ \hat{y}_{n+1}^{(k)} \in \pi^{-1}(z; \theta) \right\} + \mathbb{1}\left\{ \hat{y}_{n+1}^{(k)} \notin \pi^{-1}(z; \theta) \right\}$$

$$\quad + \mathbb{1}\left\{ \hat{y}_{n+1}^{(k)} \in \pi^{-1}(z; \theta) \right\} \cdot \left( \min_{\alpha \in [0,1)} \left\{ \alpha \,\Big|\, \mathcal{C}^{(k)}(x_{n+1}; \alpha) \subseteq \pi^{-1}(z; \theta) \right\} - 1 \right)$$

$$= 1 - \left( 1 - \min_{\alpha \in [0,1)} \left\{ \alpha \,\Big|\, \mathcal{C}^{(k)}(x_{n+1}; \alpha) \subseteq \pi^{-1}(z; \theta) \right\} \right) \cdot \mathbb{1}\left\{ \hat{y}_{n+1}^{(k)} \in \pi^{-1}(z; \theta) \right\}$$

$$= 1 - w^{(k)}(z, x_{n+1}) \cdot \mathbb{1}\left\{ \hat{y}_{n+1}^{(k)} \in \pi^{-1}(z; \theta) \right\},$$

where in the last equation, we denote the conformalized weight as:

$$w^{(k)}(z, x_{n+1}) = 1 - \min_{\alpha \in [0,1)} \left\{ \alpha \,\middle|\, \mathcal{C}^{(k)}(x_{n+1}; \alpha) \subseteq \pi^{-1}(z; \theta) \right\}.$$

The final estimator is an average over all $K$ of these estimators:

$$\hat{\alpha}(z) = 1 - \frac{1}{K} \sum_{k=1}^{K} w^{(k)}(z, x_{n+1}) \cdot \mathbb{1}\left\{ \hat{y}_{n+1}^{(k)} \in \pi^{-1}(z; \theta) \right\},$$

which concludes this part of the derivation. $\qquad\square$

*Derivation of the indicator term.* Under the linear programming assumption, there is:

$$
\begin{aligned}
\left\{ y \in \mathcal{Y} \mid y \in \pi^{-1}(z; \theta) \right\} &= \bigcap_{z' \in \mathcal{Z}(\theta)} \left\{ y \in \mathcal{Y} \mid g(z, y; \theta) \leq g(z', y; \theta) \right\} \\
&= \bigcap_{z' \in \mathcal{Z}(\theta)} \left\{ y \in \mathcal{Y} \mid g(z, y; \theta) - g(z', y; \theta) \leq 0 \right\} \\
&= \bigcap_{z' \in \mathcal{Z}(\theta)} \left\{ y \in \mathcal{Y} \mid \langle y, z \rangle - \langle y, z' \rangle \leq 0 \right\} \\
&= \bigcap_{z' \in \mathcal{Z}(\theta)} \left\{ y \in \mathcal{Y} \mid \langle y, z - z' \rangle \leq 0 \right\} \\
&= \left\{ y \in \mathcal{Y} \mid \sup_{z' \in \mathcal{Z}(\theta)} \langle y, z - z' \rangle \leq 0 \right\}.
\end{aligned}
$$

We rewrite the condition in the set as:

$$\sup_{z' \in \mathcal{Z}(\theta)} \langle y, z - z' \rangle = \langle y, z \rangle - \inf_{z' \in \mathcal{Z}(\theta)} \langle y, z' \rangle.$$

Since $\mathcal{Z}(\theta)$ is a compact set, by the Krein–Milman theorem (Krein & Milman, 1940), there is

$$\inf_{z' \in \mathcal{Z}(\theta)} \langle y, z' \rangle = \inf_{v \in \mathcal{V}(\theta)} \langle y, v \rangle.$$

where we denote $\mathcal{V}(\theta)$ as the collection of extreme points of $\mathcal{Z}(\theta)$. Plugging in the original equation, we get

$$\left\{ y \in \mathcal{Y} \mid y \in \pi^{-1}(z; \theta) \right\} = \left\{ y \in \mathcal{Y} \mid \sup_{v \in \mathcal{V}(\theta)} \langle y, z - v \rangle \leq 0 \right\} = \bigcap_{v \in \mathcal{V}(\theta)} \left\{ y \in \mathcal{Y} \mid \langle y, z - v \rangle \leq 0 \right\}.$$

Therefore, there is:

$$\mathbb{1}\left\{ \hat{y}_{n+1}^{(k)} \in \pi^{-1}(z; \theta) \right\} = \prod_{v \in \mathcal{V}(\theta)} \mathbb{1}\left\{ \langle \hat{y}_{n+1}^{(k)}, z - v \rangle \leq 0 \right\}.$$

This concludes this part of the derivation. $\qquad\square$

*Derivation of the conformalized weight.* Recall that the conformalized weight is defined as

$$w^{(k)}(z, x_{n+1}) = 1 - \min_{\alpha \in [0,1)} \left\{ \alpha \,\middle|\, \mathcal{C}^{(k)}(x_{n+1}; \alpha) \subseteq \pi^{-1}(z; \theta) \right\}.$$

By the definition of the conformalized set, there is the following equivalence:

$$\mathcal{C}^{(k)}(x_{n+1}; \alpha) \subseteq \pi^{-1}(z; \theta) \iff \hat{R}(\alpha) \leq \inf_{y \in \partial \pi^{-1}(z; \theta)} \| \hat{y}_{n+1}^{(k)} - y \|_2,$$

where the right-hand side represents the closest distance to the boundary of a $\pi^{-1}(z; \theta)$. Under linear programming assumption, the inverse feasible region can be written as

$$\pi^{-1}(z; \theta) := \bigcap_{z' \in \mathcal{Z}(\theta)} \left\{ y \in \mathbb{R}^d \mid y^\top z \leq y^\top z' \right\} = \bigcap_{v \in \mathcal{V}(\theta)} \left\{ y \in \mathbb{R}^d \mid y^\top (z - v) \leq 0 \right\},$$

Therefore, $\pi^{-1}(z;\theta)$ is a polyhedral cone, which is a group of intersecting hyperplanes. Therefore, the closest distance to the boundary of this cone can be derived as:

$$\inf_{y \in \partial \pi^{-1}(z;\theta)} \|\hat{y}_{n+1}^{(k)} - y\|_2 = \min_{v \in \mathcal{V}(\theta) \setminus \{z\}} \frac{\left| \hat{y}_{n+1}^{(k)\top}(z - v) \right|}{\|z - v\|_2} =: \hat{D}^{(k)}.$$

We denote the right-hand side quantity as $\hat{D}^{(k)}$. Therefore, by the definition of $\hat{R}(\alpha)$, there is

$$\hat{R}(\alpha) \leq \hat{D}^{(k)} \iff \alpha \geq \frac{\hat{D}^{(k)} + \sum_{i=1}^{n} \|\hat{y}_i - y_i\|_2}{(n+1)\hat{D}^{(k)}} \geq \frac{1}{n+1} \text{ and } \alpha < 1.$$

Therefore, the solution to the optimization problem yields a closed-form solution as:

$$\min_{\alpha \in [0,1)} \left\{ \alpha \mid \mathcal{C}^{(k)}(x_{n+1}; \alpha) \subseteq \pi^{-1}(z;\theta) \right\} = \min \left\{ \frac{\hat{D}^{(k)} + \sum_{i=1}^{n} \|\hat{y}_i - y_i\|_2}{(n+1)\hat{D}^{(k)}}, 1 \right\}.$$

Therefore, plugging this back to the conformalized weight:

$$w^{(k)}(z, x_{n+1}) = \left( \frac{n\hat{D}^{(k)} - \sum_{i=1}^{n} \|\hat{y}_i - y_i\|_2}{(n+1)\hat{D}^{(k)}} \right)^{+},$$

where $(\cdot)^{+}$ is the positive part operator. This concludes the proof. $\square$

## D  PROOF OF PROPOSITION 3

*Proof.* For notation simplicity, denote the sets in the numerator and the denominator of TPR defined in equation 11 as:

$$A = \{z \in \mathcal{Z}(\theta) \mid \mathbb{P}\{z \notin \pi(Y;\theta)\} < 1 \text{ and } \hat{\alpha}(z) < 1\},$$
$$B = \{z \in \mathcal{Z}(\theta) \mid \mathbb{P}\{z \notin \pi(Y;\theta)\} < 1\}.$$

Note that $A \subseteq B$. Without loss of generality, we assume that $B$ (therefore $A$) is a finite set, *i.e.*, there is only a finite set of decisions that have ground-truth risk smaller than one[4].

Since $B$ is a constant irrelevant to $K$, we begin by expanding the following expression:

$$\mathbb{E}[\#A] = \mathbb{E}\left[ \sum_{z \in B} \mathbb{1}\{\hat{\alpha}(z) < 1\} \right]$$

$$= \mathbb{E}\left[ \sum_{z \in B} \left( 1 - \prod_{k=1}^{K} \left( \mathbb{1}\left\{ \hat{y}_{n+1}^{(k)} \notin \pi^{-1}(z;\theta) \right\} \cdot \mathbb{1}\left\{ w^{(k)}(z, x_{n+1}) > 0 \right\} \right) \right) \right]$$

$$= \mathbb{E}[\#B] - \mathbb{E}\left[ \sum_{z \in B} \prod_{k=1}^{K} A_k \right].$$

Here, the second equality results from the observation that when: $(i)$ at least one of the model predictions falls within $\pi^{-1}(z;\theta)$, and $(ii)$ its conformalized weight is not zero, then the estimated risk after averaging would be less than one. In the last equality, we denote

$$A_k = \mathbb{1}\left\{ \hat{y}_{n+1}^{(k)} \notin \pi^{-1}(z;\theta) \right\} \cdot \mathbb{1}\left\{ w^{(k)}(z, x_{n+1}) > 0 \right\}.$$

Note that $A_k \in \{0,1\}$ and has a nonzero probability taking one[5]. Since at the right hand side, the term $\prod_{k=1}^{K} A_k$ monotonically decreases with $K$ almost surely, then $\mathbb{E}[\#A]$ monotonically increase with $K$. Since TPR $= \mathbb{E}[\#A]/\mathbb{E}[\#B]$, we know that TPR monotonically increases with $K$. This concludes the proof. $\square$

---

[4]This proof naturally extends to the infinite case by replacing the counting measure "#" with continuous measures, such as Lebesgue measure defined within the decision space $\mathcal{Z}$.

[5]Under some regularity condition on the distribution of $Y$, *e.g.*, is supported on the whole space $\mathcal{Y}$. Without this condition, the statement also holds by changing "monotonically increasing" to "non-decreasing".

# E    EXPERIMENT SETTING

This section is organized as follows: Appendix E.1 shows the computation resources used in our experiments. Appendix E.2 and Appendix E.3 presents the detailed configurations for the two synthetic settings and the real-world case study, respectively. Appendix E.4 describes the detailed configurations of CREDO, the baselines, and the procedures for the two experiments (Conservativeness versus Accuracy Tradeoff and Decision Quality Evaluation).

## E.1    COMPUTATION RESOURCES

Table 2: Computation resources specfications

| Operating System | CPU | RAM | GPU |
|---|---|---|---|
| Windows 11 | 13th-generation Intel Core i7, 16 cores | 16GB | Not used |

The computational resources used in our experiments are detailed in Table 2. Our code is implemented in Python, with key dependencies including Scikit-learn (Pedregosa et al., 2011) for solving linear programs and working with Gaussian mixture models (fitting and sampling), and CDD (Fukuda, 1997) for computing the vertices of polytopes. A complete list of dependencies and their version numbers is available in our codebase.

In terms of execution time, each single experiment (consisting of multiple independent trials) in the paper takes less than 30 minutes to run. All models can be executed almost instantly for a single trial in our experiment, except SPO+ (Elmachtoub & Grigas, 2022). The main computational bottleneck of SPO+ (Elmachtoub & Grigas, 2022) arises from the need to solve two linear programming problems for each iteration of parameter update, which can be easily addressed by using more efficient solvers.

## E.2    SYNTHETIC SETTING

We denote $\sigma$ as the component variance scale ($\sigma = 1$ by default), and we denote $\mathbf{I}_2$ as the two-dimensional identity matrix.

**Setting I**    A linear programming problem featuring a triangular feasible region with three vertex decisions, defined as:

$$\max_{z \in \mathbb{R}^2} \{Y_1 z_1 + Y_2 z_2 \mid z_1 + z_2 \leq 1, z_i \geq 0, z_2 \geq 0\}, \quad \begin{pmatrix} Y_1 \\ Y_2 \end{pmatrix} \sim \mathcal{N}\left(\begin{pmatrix} -1 \\ -1 \end{pmatrix}, \sigma \cdot \mathbf{I}_2\right).$$

This optimization problem can be interpreted as a profit maximization task, where a manufacturer chooses the optimal production quantities $z_1$ and $z_2$ under a budget constraint. The Gaussian random revenues $Y_1$ and $Y_2$ have negative expected values but may exhibit some variance, capturing a risky market scenario that could still yield profit under favorable conditions.

The feasible region in this problem can be more compactly denoted by its constraint matrix $\mathbf{A}$ and constraint vector $\mathbf{b}$ as

$$\mathbf{A} = \begin{pmatrix} 1 & 1 \\ -1 & 0 \\ 0 & -1 \end{pmatrix} \quad \mathbf{b} = \begin{pmatrix} 1 \\ 0 \\ 0 \end{pmatrix}.$$

**Setting II**    A linear programming problem that employs a more complex octagonal feasible region with five vertices and multimodal objective uncertainty, allowing us to assess performance in scenarios with multiple potentially optimal decisions. Specifically, the optimization problem is set up as the canonical form $\max_{z \in \mathbb{R}^2} \{Y^\top z \mid \mathbf{A}z \leq \mathbf{b}\}$, where the constraint matrices $\mathbf{A}$ and constraint vector

$\mathbf{b}$ are defined as

$$
\mathbf{A} = \begin{pmatrix} -0.5 & -1 \\ 0 & -1 \\ -0.5 & 1 \\ 0.5 & 1 \\ 2 & -1 \\ 1 & 0 \\ 0 & 1 \\ -1 & 0 \end{pmatrix} \quad \mathbf{b} = \begin{pmatrix} -1 \\ 0 \\ 1 \\ 5 \\ 10 \\ 5.5 \\ 2.5 \\ -1 \end{pmatrix}.
$$

The random vector $Y \in \mathbb{R}^2$ is drawn from a three-component Gaussian mixture distribution

$$
p(x) = \sum_{k=1}^{3} w_k \mathcal{N}\left( x \mid \mu_k, \ \sigma \cdot \sigma_k^2 \mathbf{I}_2 \right),
$$

where the mixture weights are $\boldsymbol{w} = (0.3, 0.4, 0.3)$, the component means are

$$
\mu_1 = \begin{pmatrix} 0.0 \\ -0.8 \end{pmatrix}, \quad \mu_2 = \begin{pmatrix} -0.5 \\ 0.25 \end{pmatrix}, \quad \mu_3 = \begin{pmatrix} 0.8 \\ -0.1 \end{pmatrix},
$$

and the component variances are

$$
\sigma_1^2 = (0.01)^2, \quad \sigma_2^2 = (0.03)^2, \quad \sigma_3^2 = (0.02)^2.
$$

### E.3 REAL-WORLD SETTING

As described in the main paper, we consider a real-world power grid investment decision-making problem (Zhou et al., 2024). We begin by introducing the application background, followed by a formalization of the problem as a knapsack optimization. Finally, we present a linear programming relaxation of this problem, which can be directly used in `CREDO`.

**Background** A utility company based in Indianapolis, Indiana, has compiled detailed records of over 1,700 solar panel installations between 2010 and 2024, including the installation dates and affiliated grid components. With the renewable energy sector now at full scale, the management team anticipates a steady and significant monthly increase in solar adoption in the downtown area. In preparation for the incoming demand, they are planning targeted upgrades to grid-level inverters at four selected substations (we refer to them as Substation A to D) under a limited budget. The utility company would like to consult on `CREDO`'s suggestion for a list of candidate upgrade plans that would most likely be optimal.

The data is available in our codebase in a spatio-temporally aggregated format (monthly, by substation), with substation names anonymized. However, the granular solar panel installation data used in this study cannot be shared publicly, as it is proprietary to the utility company.

**Mathematical Formulation** This problem can be mathematically formulated as a penalized knapsack problem, whose goal is to minimize the total penalty from capacity violations while ensuring that the total upgrade cost does not exceed the budget. Formally, define $d$ as the total number of substations, and define parameters as in Table 3, the optimization problem can be written as

$$
\min_{\mathbf{a} \in \{0,1\}^d} \left\{ \sum_{i=1}^{d} \mathbb{1}(Y_i \geq \tau_i)(1 - a_i) l_i \ \Big| \ \sum_{i=1}^{d} a_i c_i \leq b \right\}. \tag{18}
$$

In our experiment, we configure these parameters as follows: we assume the cost $c_i$ and loss $l_i$ for all substations equals one unit. The capacity threshold $\tau_i$ is set to be the historical average solar panel monthly increment. The budget $b$ is set to half the cost of upgrading all substations, allowing at most two out of four substations can be upgraded. Note that these parameters require additional information from the company, which is not available at the time of writing this paper.

Table 3: Parameter definitions for the real-world optimization problem

| Parameter | Definition |
|-----------|------------|
| $a_i$ | Binary decision variable; $a_i = 1$ indicates a substation upgrade, and $a_i = 0$ indicates no upgrade. |
| $Y_i$ | Random variable representing the monthly increase in solar panel installations. |
| $c_i$ | Cost associated with upgrading substation $i$. |
| $\tau_i$ | Capacity threshold—the maximum number of allowable solar connections at substation $i$. |
| $l_i$ | Penalty incurred if the threshold $\tau_i$ is exceeded without an upgrade. |
| $b$ | Total budget. |

**Relaxed Formulation**   The knapsack problem defined in equation 18 can be relaxed into the following linear programming problem:

$$\min_{z \in [0,1]^d} \left\{ \sum_{i=1}^{d} \frac{l_i}{1 + e^{-\beta(Y_i - \tau_i)}} (1 - a_i) \ \middle| \ -\sum_{i=1}^{d} c_i z_i \leq b - \sum_{i=1}^{d} c_i \right\}, \tag{19}$$

where $\beta$ is an arbitrary smoothing parameter that we set $\beta = 0.5$. Note that in the equation above, two relaxations have been applied:

- The binary random variable $\mathbb{1}\{Y_i \geq \tau_i\}$ has been relaxed to a continuous random variable over $[0,1]$ using the sigmoid function, where the smoothing parameter $\beta$ controls the sharpness of the approximation.
- The decision space of $a_i$ has been relaxed from the discrete set $0,1^d$ to the continuous unit hypercube $[0,1]^d$. This relaxation does not introduce approximation bias, as the optimal solutions to the original and relaxed problems are almost surely the same – both attained at one of the hypercube's vertices.

Denote the vectors

$$Y = \begin{pmatrix} \frac{l_i}{1 + e^{-\beta(Y_i - \tau_i)}} \\ \ldots \\ \frac{l_i}{1 + e^{-\beta(Y_i - \tau_i)}} \end{pmatrix}, \quad \mathbf{c} = \begin{pmatrix} c_1 \\ \ldots \\ c_n \end{pmatrix}, \quad z = \begin{pmatrix} 1 - a_1 \\ \ldots \\ 1 - a_n \end{pmatrix}, \tag{20}$$

we can compactly write the optimization problem above as

$$\min_{z \in \mathbb{R}^d} \left\{ Y^\top z \mid -\mathbf{c}^\top z \leq b - \mathbf{1}_d^\top \mathbf{c}, \ \mathbf{0} \leq z \leq \mathbf{1} \right\}. \tag{21}$$

This can be written as the canonical form of linear programming $\min_{z \in \mathbb{R}^d} \left\{ Y^\top z \mid \mathbf{A} z \leq \mathbf{b} \right\}$ by defining

$$\mathbf{A} = \begin{pmatrix} -\mathbf{c}^\top \\ \mathbf{I}_d \\ -\mathbf{I}_d \end{pmatrix}, \quad \mathbf{b} = \begin{pmatrix} b - \mathbf{1}_d^\top \mathbf{c} \\ \mathbf{1}_d \\ \mathbf{0}_d \end{pmatrix},$$

where $\mathbf{I}_d$ denotes the $d$-dimensional identity matrix, and $\mathbf{1}_d$ and $\mathbf{0}_d$ denote the $d$-dimensional all-ones and all-zeros vectors, respectively. These parameters can be directly input to the CREDO framework using the data $Y$ obtained via sigmoid transformation defined in equation 20 to obtain decision risk estimates.

Therefore, with the above reformulation in equation 21, the problem is now a linear programming problem, where the closed-form risk estimator derived in Corollary 1 can be directly applied.

### E.4   BENCHMARKING BASELINES DESCRIPTION

In this part, we describe the p-value conformalized radius, baselines, metrics, as well as the detailed evaluation procedure for each experiment that we conducted in the main paper. For both experiments, the following (Table 4) default values of hyperparameters are used for CREDO.

The p-value conformalized radius (Singh et al., 2024) originates from the standard conformal prediction literature, defined as:

$$\hat{R}(\alpha) = \hat{Q}\left(\frac{\lceil (n+1)(1-\alpha) \rceil}{n}\right)$$

where $\hat{Q}(\cdot)$ denotes the empirical quantile function of the calibrated nonconformity scores $\{\|\hat{y}_i - y_i\|_2\}_{i=1}^n$, with $\hat{y}_i$ being a generated prediction from $\hat{f}(x_i)$[6].

Table 4: Hyperparameters in the Convservativeness vs. Accuracy Tradeoff experiment.

| Hyperparameter | Description | Default value |
|:---:|:---|:---|
| $K$ | Simulation size | $1 \times 10^2$ |
| $\sigma$ | (Component) variance scale | 1 |
| $n$ | Calibration dataset size | $1 \times 10^2$ |
| $m$ | Training dataset size | $1 \times 10^2$ |

**Convservativeness vs. Accuracy Tradeoff**    Table 5 summarizes the used baslines, which are two ablation models mainly target replacing step 2 of `CREDO` with different methods that can evaluate the probability $\mathbb{P}\left\{Y \in \pi^{-1}(z; \theta)\right\}$. We note that we implement `Point` by setting `CREDO` with $K = 1$ and configure the probabilistic model to collapse to a point mass centered at the mean.

Table 5: Summary table of baselines used in the Convservativeness vs. Accuracy Tradeoff experiment

| Model Name | Description | Additional Remark |
|:---:|:---|:---|
| `Point` | A deterministic variant of `CREDO`, where the generative model $\hat{f}$ is replaced by a deterministic point prediction model. The prediction model captures the mean of the distribution of $Y$. Note that this ablation model is adopted from previous literature (Singh et al., 2024), where it was originally applied to assessing the risk of given prediction intervals. | The design of `Point` limits its ability to capture multimodal or heteroskedastic uncertainty in the objective coefficient space. |
| `NS` | A naive sampling estimator computed as $$\hat{\alpha}_{\text{NS}}(z) = 1 - \frac{1}{K}\sum_{k=1}^{K} \mathbb{1}\left\{\hat{y}_{n+1}^{(k)} \in \pi^{-1}(z; \theta)\right\},$$ where $\hat{y}_{n+1}^{(1)}, \ldots \hat{y}_{n+1}^{(K)}$ are independent samples drawn from $\hat{f}(x_{n+1})$. This estimator directly approximates the probability mass over the inverse feasible region using samples from the generative model. | The design of `NS` contrasts the proposed estimator, which can be rewritten as equation 10, by dropping the conformalized weighting term. This limits its ability to meet the conservativeness guarantee. |

Table 6 summarizes the evaluation metrics used in this experiment. We use $T$ to denote the total number of repeated independent trials, where we set $T = 20$ across all of our experiments. For each metric, the error is reported as the *standard error of the mean (SEM)*, calculated as the standard deviation of the metric values across trials divided by $\sqrt{T}$. Note that while most metrics require summing over all feasible decisions, we can simplify to summing all the decisions defined on the vertices of the feasible region, as we have justified in the main text.

**Decision Quality Evaluation**    Table 7 presents the four selected baselines used in the Decision Quality Evaluation experiment. The baselines have been selected to satisfy the following criteria: (i) the ability to produce a decision that maximizes a (linear) objective function; and (ii) the capacity to handle randomness in the underlying optimization problem.

---

[6]We set $\hat{Q}(0) = 0$ and $\hat{Q}(1) = +\infty$ to complete the definition.

Table 6: Summary table of metrics used in the Convservativeness vs. Accuracy Tradeoff experiment.

| Metric Name | Description | Mathematical Formula |
|---|---|---|
| Conservativeness rate | The percentage of decisions where the risk estimate satisfies the conservativeness guarantee in equation 2 averaged across all trails. | $\frac{1}{T \cdot \|\mathcal{V}(\theta)\|} \sum_{z \in \mathcal{V}(\theta)} \mathbb{1}\{\hat{\alpha}(z) \leq \alpha(z)\}$ |
| True positive rate | The percentage of decisions where the method correctly identifies them to have risk of less than one, averaged across all trials. | $\frac{1}{T \cdot \|\mathcal{V}(\theta)\|} \sum_{z \in \mathcal{V}(\theta)} \mathbb{1}\{\hat{\alpha}(z) < 0\} \cdot \mathbb{1}\{\alpha(z) < 0\}$ |
| Relative accuracy | The negative absolute differences between the estimated and true risks across all decisions and trials, normalized to $[0, 1]$ across different baselines. | $-\frac{1}{T \cdot \|\mathcal{V}(\theta)\|} \sum_{z \in \mathcal{V}(\theta)} \|\hat{\alpha}(z) - \alpha(z)\|$ |

Table 7: Summary table of baselines used in the Decision Quality Evaluation experiment

| Model Name | Description | Additional remarks |
|---|---|---|
| PTO | The standard two-stage predict-then-optimize approach (Bertsimas & Kallus, 2020), which first predicts parameters and then solves the resulting optimization problem; $$\min_{z \in \mathcal{Z}(\theta)} g(z, \hat{Y}),$$ where $\hat{Y}$ is a point estimate of $\mathbb{E}[Y\|X]$. | Since $X$ is omitted in our setting, PTO degenerates to stochastic optimization, and $\hat{Y}$ can be simply specified as the empirical average estimator. |
| RO | Robust optimization defined as $$\min_{z \in \mathcal{Z}(\theta)} \max_{y \in \mathcal{U}} g(z, y)$$ where $\mathcal{U} \subseteq \mathcal{Y}$ is the uncertainty set of $Y$. | The uncertainty set is constructed in a data-driven manner by applying naive conformal prediction to $Y$, using the $\ell_\infty$ norm residual as the nonconformity score (Zhou et al., 2024). |
| SPO+ | A predict-then-optimize method where the prediction model is trained using the surrogate smart predict-then-optimize loss (Elmachtoub & Grigas, 2022) | For each trial, it is trained for $1 \times 10^2$ epochs using a learning rate of $1 \times 10^{-1}$. |
| DFL | Decision-focused learning, where the prediction model is trained by directly optimizing the downstream objective through end-to-end differentiation of the optimization layer (Amos & Kolter, 2017). | For each trial, it is trained for $1 \times 10^2$ epochs using a learning rate of $1 \times 10^{-1}$. |

As for the evaluation metric, Algorithm 3 outlines the generic procedure for computing empirical confidence rankings in a single trial. In the synthetic setting, this procedure is repeated over $T = 1 \times 10^2$ independent trials with $\ell = 1 \times 10^3$ test data points. Meanwhile, in the real-world setting, the evaluation is performed rolling over $T = 12$ periods, spanning from 2010 to 2022. Each trial uses a two-year window (24 months) of data, which is sequentially split into training, calibration, and testing sets in an $8:8:8$ ratio. The reported error is the *standard error of the mean (SEM)*, computed as the standard deviation of the metric across trials divided by $\sqrt{T}$.

---

**Algorithm 3** Empirical Confidence Ranking

---

**Require:** Original optimization $\pi^* : \mathcal{Y} \to \mathcal{Z}$; Evaluated policy $\pi : \mathcal{X} \to \mathcal{Z}$; Test dataset $\{(x_i, y_i)\}_{i=1}^{\ell}$.
1: $\mathcal{Z}^* = \emptyset$
2: **for** $i = \{1, \ldots, \ell\}$ **do**
3:     $z_i^* \leftarrow \pi^*(y_i)$.
4:     $\mathcal{Z}^* \leftarrow \mathcal{Z}^* \cup \{z_i^*\}$.
5: **end for**
6: **for** $z \in \mathcal{V}(\theta)$ **do**
7:     $h(z) \leftarrow$ the total occurence of decision $|\{i \mid z_i^* = z\}|$.
8: **end for**
9: **for** $i = \{1, \ldots, \ell\}$ **do**
10:     $\hat{z}_i \leftarrow \pi(x_i)$.
11:     $\text{rank}_i \leftarrow$ ranking of estimated decision $|\{z' \in \mathcal{V}(\theta) \mid h(z') \geq h(\hat{z}_i)\}|$
12: **end for**
13: **return** Empirical confidence ranking $1/\ell \cdot \sum_{i=1}^{\ell} \text{rank}_i$.

---

## F   Additional experiment results

This section shows additional experiments from the two experiments (Conservativeness vs. Accuracy Tradeoff, Decision Quality Evaluation) that were shown in the main paper, as well as an additional experiment that investigates the effect of misspecified models on the risk estimation under the `CREDO` framework.

**Conservativesness vs. Accuracy Tradeoff**   In this section, we first present three additional insights based on the results in Figure 5, followed by the complete set of experimental results that were only partially shown in the main paper. These experiments aim to investigate the sensitivity of our method's accuracy to different hyperparameter settings. At a high level, we find that `CREDO` consistently outperforms `Point`, with the performance gap becoming more pronounced at larger variance scales $\sigma$ and higher simulation sizes $K$.

From Figure 5, we draw the following observations in addition to those discussed in the main paper:

- A general trend observed in both settings is the decline in relative efficiency. This is expected: conformal prediction must adopt a more conservative approach to preserve its validity guarantees as the data becomes more stochastic. While this ensures coverage, it also results in wider prediction sets and, consequently, lower relative efficiency.

- In the first example, `PointModel` achieves the best performance when the variance is small. This is because the generated distribution is tightly concentrated around the origin $(0,0)$, rendering the problem effectively deterministic—an ideal scenario for point estimation. In contrast, the second example features mixture components that are inherently dispersed across different inverse optimality regions. As a result, the data remains non-deterministic even at low variance levels, and `CREDO` consistently outperforms the baselines by capturing this underlying distributional uncertainty.

- The performance gap between `NS` and `CREDO` leads to an alternative explanation of our proposed method: the weighting term takes value from 0 to 1, and "safeguards" the estimation from being overestimation by downscaling the contribution of each sample that would fall within region $\pi^{-1}(z)$. The value of this weight is determined through a data-driven, conformalized procedure, ensuring conservativeness is achieved. This proves the significance of our proposed algorithm.

Figure 6, Figure 7, Figure 8, Figure 9 illustrate the full results evaluating the accuracy of three methods under three hyperparameter selections. We make two remarks:

- Observe that `NS` achieves the highest accuracy metrics across nearly all plots. This outcome is expected, as `NS` is explicitly designed to maximize accuracy. However, it fails to satisfy the conservativeness guarantee, rendering its performance substantially less reliable in

practice. Therefore, we exclude it from the result demonstration in our experiment result comparisons.

- Observe that CREDO consistently achieves higher accuracy metrics than Point, highlighting the advantage of incorporating generative models in CREDO over relying solely on point predictions.

Additionally, Figure 10 and Figure 11 present the raw estimated risks for all decisions across the three methods under Setting I and Setting II. These plots provide a more fundamental perspective underlying the summarized metrics shown previously. Each boxplot is computed over 20 independent trials using the default hyperparameter settings. CREDO and Point provide confidence estimates that fall below it, NS may overshoot to the red region. This indicates the conservativeness guarantee provided by the conformailized procedure in our framework.

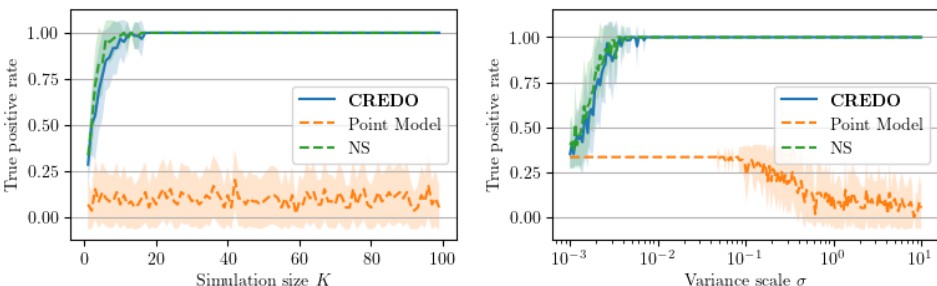

Figure 6: True positive rate versus simulation size $K$ (left) and varaince scale $\sigma$ (right) in Setting I.

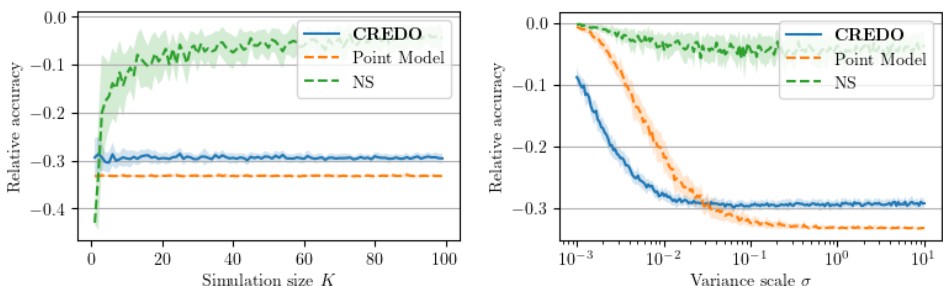

Figure 7: Relative accuracy versus simulation size $K$ (left) and varaince scale $\sigma$ (right) in Setting I.

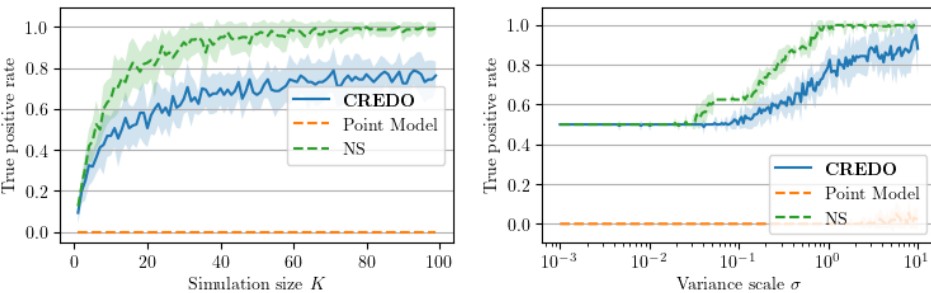

Figure 8: True positive rate versus simulation size $K$ (left) and varaince scale $\sigma$ (right) in Setting II.

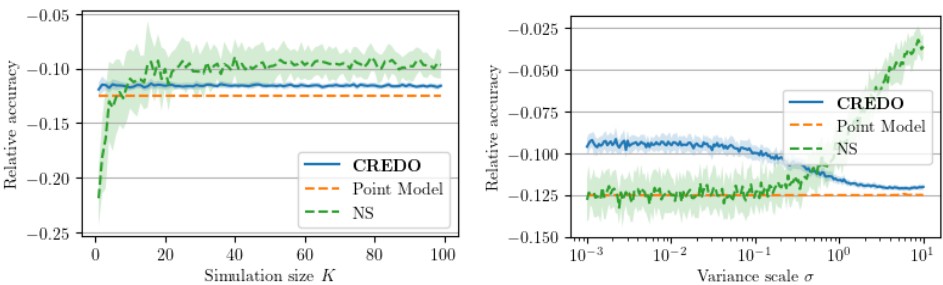

Figure 9: Relative accuracy versus simulation size $K$ (left) and varaince scale $\sigma$ (right) in Setting II.

**Decision Quality Evaluation**   Figure 12 shows the frequency that the decisions get picked for each baseline of Setting I and Setting II. In both settings, the output of CREDO closely aligns with the `Simulation` benchmark computed from the ground-truth distribution, whereas most other baselines tend to select different decisions. This highlights two key points: (i) the maximum likelihood decision rule can lead to substantially different decisions compared to methods that optimize the objective without accounting for probabilistic structure; and (ii) in such a case, CREDO can provides a reliable first-stage identification of the magnitude of risk for each provided decision.

Figure 13 presents the top four candidate upgrade plans identified by CREDO. The lowest-risk, and thus most confident, recommendation is to upgrade substations A and C. This highlights CREDO's prioritization of these two substations for immediate upgrades in future operations. While the second- to fourth-ranked combinations may not be the most likely to be optimal under the model's risk assessment, they may still be valuable in practice. These alternatives might offer advantages along other dimensions not captured by the knapsack formulation, such as geographic equity, policy constraints, or logistical feasibility. We envision CREDO as a decision support tool that complements, rather than replaces, human judgment. Its ranked recommendations provide a diverse set of plausible solutions, enabling informed and context-aware decision-making instead of enforcing a single deterministic outcome.

**Additional experiment: Misspecified Model**   We conduct an additional experiment to evaluate the conservativeness–accuracy tradeoff of various models under a setting where the prediction or generative model is entirely misspecified. Our results show that, despite the misspecification, CREDO consistently maintains conservative estimates while achieving relatively high efficiency compared to its point prediction variant.

Specifically, the data generation distribution and model distribution are assumed to take the following form

$$\text{GM}_{\text{real}} = 0.4 \cdot \mathcal{N}\left(\begin{pmatrix}-0.5\\0\end{pmatrix}, \begin{pmatrix}\sigma & 0\\0 & \sigma\end{pmatrix}\right) + 0.6 \cdot \mathcal{N}\left(\begin{pmatrix}0.5\\0\end{pmatrix}, \begin{pmatrix}\sigma & 0\\0 & \sigma\end{pmatrix}\right);$$

$$\text{GM}_{\text{model}} = 0.6 \cdot \mathcal{N}\left(\begin{pmatrix}-0.5\\0\end{pmatrix}, \begin{pmatrix}\sigma & 0\\0 & \sigma\end{pmatrix}\right) + 0.4 \cdot \mathcal{N}\left(\begin{pmatrix}0.5\\0\end{pmatrix}, \begin{pmatrix}\sigma & 0\\0 & \sigma\end{pmatrix}\right).$$

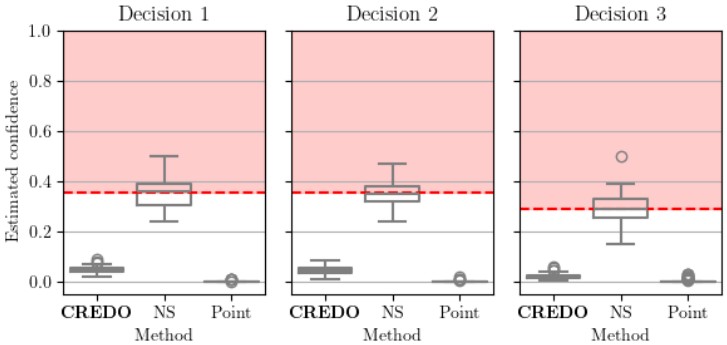

Figure 10: Estimated confidence level $(1 - \alpha)$ versus different baseline models in Setting I. The red dashed line indicates actual confidence computed from the ground-truth distribution.

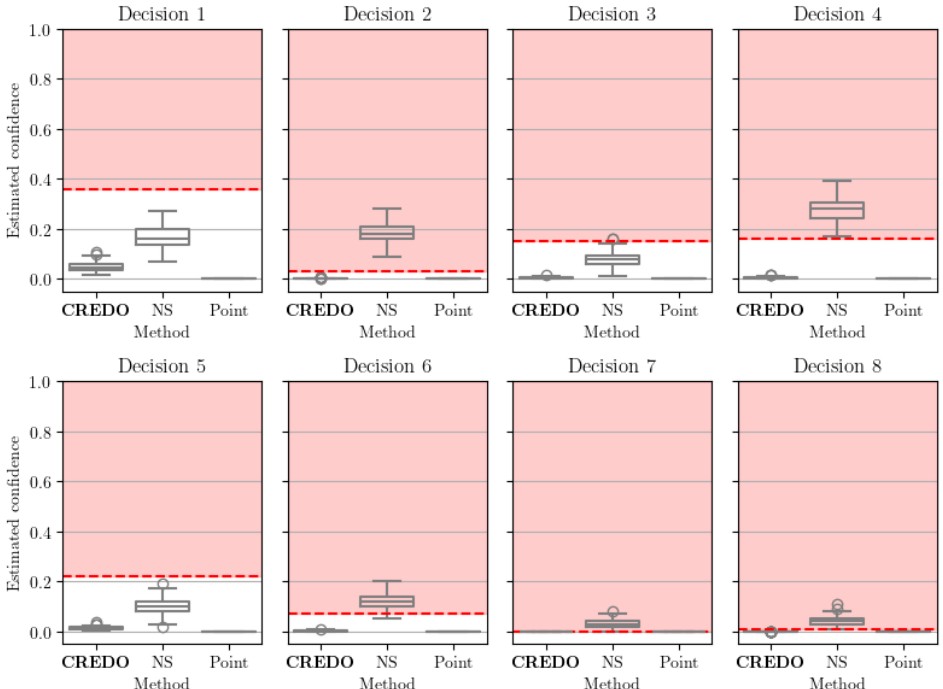

Figure 11: Estimated confidence level $(1 - \alpha)$ versus different baseline models in Setting II. The red dashed line indicates actual confidence computed from the ground-truth distribution.

Figure 14 provides the raw view of estimated risk versus the different specifications of calibration dataset size under Setting I. Figure 15 highlights the inverse feasible region correspondence from each decision in the feasible region.

Figure 16 presents the main results of this experiment, illustrating the estimated confidence levels of different models across various decisions. Due to model misspecification, NS notably overestimates the confidence for Decision 2. In contrast, both CREDO and Point avoid such overestimation, aligning with their conservativeness guarantees. Furthermore, across all decisions, CREDO consistently yields higher confidence than Point, indicating superior accuracy.

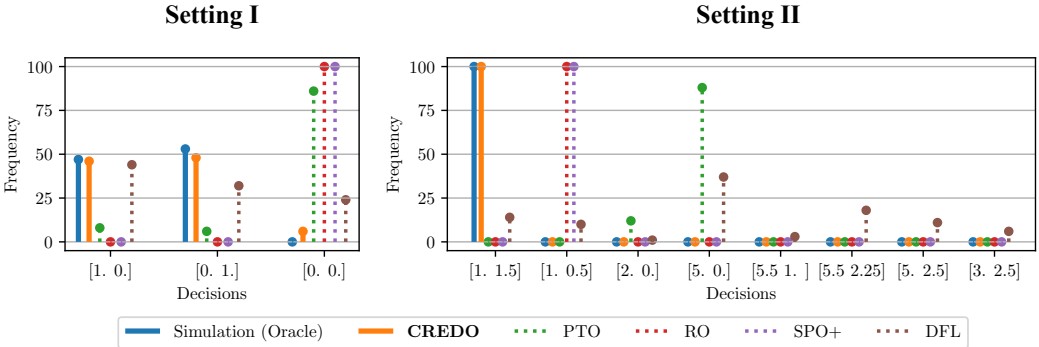

Figure 12: Frequency of each selected decision over 100 repeated trials, compared across all baseline methods. The left panel corresponds to Setting I, and the right to Setting II.

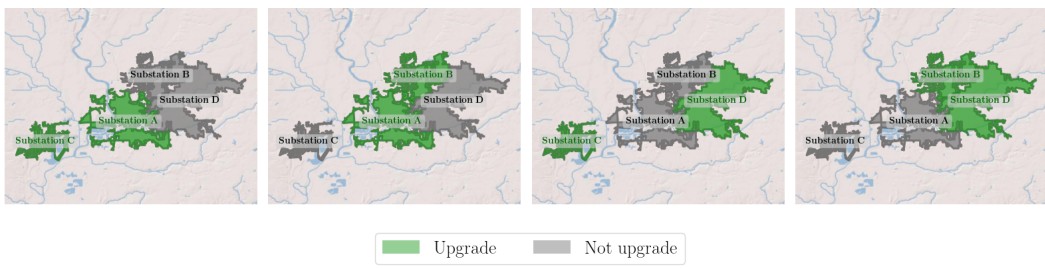

Figure 13: Top four candidate upgrade decisions with the lowest estimated risks (left to right), as recommended by CREDO. Each shaded region represents the span of a substation network.

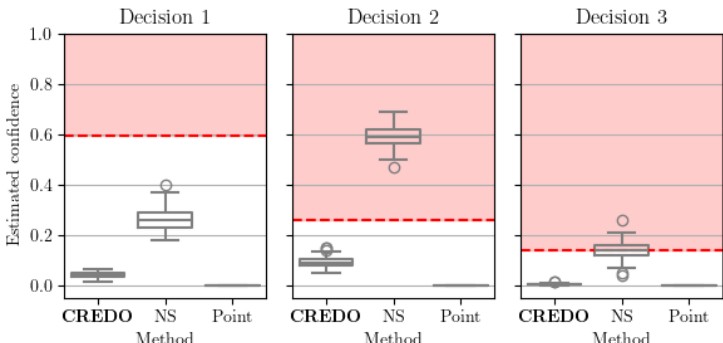

Figure 16: Estimated confidence level $(1 - \alpha)$ versus different baseline models in the misspecified model setting. The red dashed line indicates actual confidence computed from the ground-truth distribution.

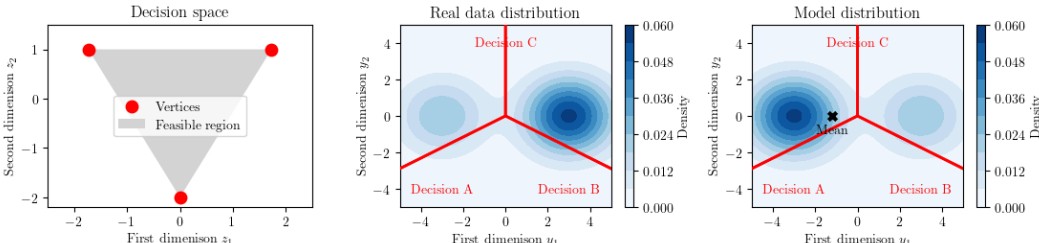

Figure 14: Setting visualization of the misspecified model experiment setting

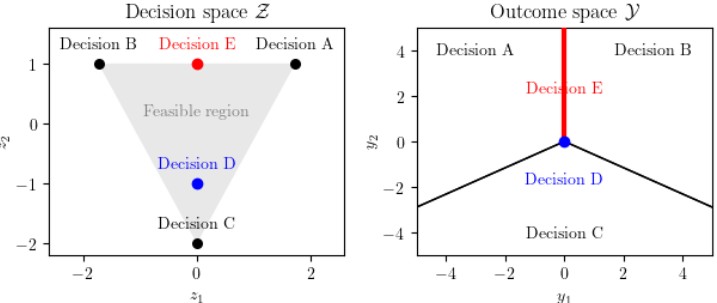

Figure 15: Decisions space and outcome space correspondence through the inverse feasible mapping in our misspecified experiment problem setting.

