# OpenReview forum: "Conformalized Decision Risk Assessment"
_ICLR.cc/2026/Conference — ICLR 2026 Poster_

### Official Review · Reviewer_pzmF · 2025-10-31

**Soundness:** 3
**Presentation:** 3
**Contribution:** 3
**Rating:** 4
**Confidence:** 4

**Summary:**

This paper proposes a new framework that can evaluate the probability of suboptimality for any decision with strong statistical guarantees. The authors reformulate the probability of suboptimality as the probability of the outcome variable belonging to an inverse feasible region. This probability can then be estimated using conformal prediction sets of varying levels of marginal coverage produced from $K$ samples from a generative model. The method is validated on two synthetic datasets and a real-world infrastructure planning problem.

**Strengths:**

The paper is well-written and well-organized. The theoretical contributions are meaningful in validating the proposed method. For example, Proposition 3 justified the use of generative models very well. This framework is a novel approach to handling risk in decision-making by quantifying it directly rather than being robust to it during optimization.

**Weaknesses:**

The authors compare their method with other robust decision-making methods using empirical confidence ranking. However, I am not convinced that this is the best metric to evaluate decision quality. How is the predicted decision’s rank in terms of its frequency in the ground truth optimal decision set informative of decision quality? This metric is instance-dependent, and so choosing an action $z$ that is optimal (i.e., $z \in \pi(Y; \theta)$), but rare among the optimal set of actions in the test set, would be discouraged by this metric. That doesn’t seem like a fair assessment of decision quality. An experiment evaluating decision quality seems crucial in building a case for viewing robust decision-making through a different lens. If the authors can clearly and strongly justify this metric or reproduce this experiment with a metric more indicative of decision quality, I will reconsider my score.

Additionally, I believe that Kiyani et al. (2025) seems quite relevant to this line of work; however, it wasn’t included in the experiments. I believe adding this baseline can strengthen the paper.

_References_
* Kiyani et al. (2025), Decision theoretic foundations for conformal prediction: Optimal uncertainty quantification for risk-averse agents. https://arxiv.org/pdf/2502.02561.

**Questions:**

* In Figure 5 Column 2, why isn’t the Point Model just a flat line? It shouldn’t be changing with $K$ increasing.
* Why isn’t NS included in Columns 2 and 3 of Figure 5?
* Column 3 of Figure 5 appears to be inaccurately interpreted (Lines 432-437). While high accuracy, in the typical sense, indicates better performance, “accuracy”, as defined by the authors, seems to be like a loss (absolute difference between true and estimated risk). So, shouldn’t the method with lower “accuracy” be better?

---

> ### Author Response · Authors · 2025-11-19
>
> We genuinely appreciate Reviewer pzmF's comments and questions. We are glad to know from the reviewer's endorsement of the method, quality, and novelty of our work. We are very happy to clarify your concerns and engage in further discussions.
>
> ## W1: Justifying empirical confidence ranking
>
> We really appreciate the reviewer for asking for clarifications. We fully agree with the reviewer that a metric that accurately reflects decision quality is crucial. Below, we gently justify why empirical confidence ranking is a good and fair metric for decision quality in our setting, and we welcome any further questions and are happy to engage in deeper discussions and/or conduct additional experiments.
>
> We would first like to gently clarify the nature of this metric.
> As the reviewer correctly pointed out, this metric is a population-level measure: it is instance-independent and would discourage decisions that are rare in the optimal set of actions in the test set (i.e., discourages decisions that are rarely optimal over simulated realizations of $Y$ in the test set).
> This is consistent with what a decision-maker would desire for a prescribed decision to achieve when operating under ex-ante uncertainty: when outcomes are not yet realized, they would prefer actions that perform reliably across plausible scenarios, not ones that happen to be optimal only in certain cherry-picked instances.
> This principle broadly applies to existing metrics for decision quality evaluation, such as expected regret, risk-averse objectives (e.g., CVaR or variance-penalized expectations), and stochastic dominance criteria, which are all instance-independent and evaluated on the test set in a way similar to the empirical confidence ranking.
>
> Next, we highlight some nice heuristic properties and unique advantages the metric offers for describing decision quality. First, empirical confidence ranking captures a fundamental aspect of decision quality that decision-makers care about: whether a prescribed decision consistently ranks among the top choices in the long run.
> Second, this ranking remains meaningful with this explanation even when the optimization problem is a stylized model, where the magnitude of the objective value may have a direct physical meaning, so it fits well in synthetic experiments.
> Third, this metric is a proper metric for all competing baselines in our experiments, since under the ground-truth condition of perfect modeling with no uncertainty in $Y$, all methods would converge to prescribing the same top-ranked (i.e., most confident) decision. Finally, we find this metric may also be favored in assessing decision quality under routine-oriented principles such as business-as-usual (BAU) analyses, as ranking-based metrics are generally numerically more robust to extreme outliers in $Y$ or objective values that might significantly affect the evaluation result. BAU is commonly adopted in grid infrastructure planning, which is an applicational setting considered in our paper.
>
> Additionally, to show the broad applicability of CREDO, we also provide an additional experiment with expected regret, which is a well-known general decision quality evaluation metric. Here, we set CREDO to prescribe decisions by choosing the decision with the minimum estimated weighted by the sample average objective values. The full results are shown in the table below. It can be seen that CREDO still consistently ranks in the top positions among the baselines under this expected regret metric. This illustrates the flexibility of CREDO as a decision co-pilot, where its risk estimates can be inputted to different decision rules to prescribe high-quality decisions that suit different metrics and decision quality correspondingly.
>
> | Method | Syn 1 $(\sigma = 1)$ | Syn 1 $(\sigma = 10)$ | Syn 1 $(\sigma = 100)$ | Syn 2 $(\sigma = 1)$ | Syn 2 $(\sigma = 10)$ | Syn 2 $(\sigma = 100)$ |
> |--------|------------------------|-------------------------|--------------------------|------------------------|-------------------------|--------------------------|
> | **PTO**   | **0.15 ± 0.01** | 0.62 ± 0.04 | 2.16 ± 0.08 | **1.07 ± 0.06** | **1.11 ± 0.06** | `1.47 ± 0.08` |
> | **RO**    | **0.15 ± 0.01**  | **0.61 ± 0.02** | **2.08 ± 0.07** | 1.30 ± 0.05 | 1.34 ± 0.05 | 1.67 ± 0.06 |
> | **SPO+**  | **0.15 ± 0.01**  | 0.63 ± 0.04 | 2.15 ± 0.08 | 1.24 ± 0.13 | 1.28 ± 0.12 | **1.45 ± 0.07** |
> | **DFL**   | 0.24 ± 0.03 | 0.68 ± 0.05 | 2.15 ± 0.09 | 1.30 ± 0.19 | 1.32 ± 0.19 | 1.69 ± 0.18 |
> ||
> | **CREDO** | **0.15 ± 0.01** | **0.61 ± 0.02** | **2.08 ± 0.07** | `1.20 ± 0.03` | `1.24 ± 0.04` | 1.67 ± 0.07 |
>
> Table: Evaluation result of expected regret for different decision-making baselines. **Bolded** font indicates the best performance, while `code-style` indicates the second-best performance.

---

> > ### Author Response · Authors · 2025-11-19
> >
> > ## W2: Related work of Kiyani et al. (2025)
> >
> > We really appreciate the reviewer for the suggestion to help strengthen our paper. We are fully aware of Kiyani et al. (2025) as an important related work when writing the current manuscript, and we have properly cited it. As suggested by the reviewer, we will further incorporate an extended discussion and comparison between Kiyani et al. (2025) and our work in the revised paper.
> >
> > Kiyani et al. (2025) examine the foundational role of uncertainty quantification in decision-making for risk-averse agents. They show that when a decision-maker evaluates decision quality using a risk-averse criterion such as VaR, the natural form of uncertainty quantification is a prediction set rather than a calibrated predictive distribution. Their framework reveals a direct connection with RO: when the uncertainty set is given by a prediction set (e.g., produced by conformal prediction), the miscoverage level of the prediction set corresponds exactly to the decision-maker’s target quantile. In this sense, the coverage level of the conformal set determines the degree of risk-aversion, and the resulting max-min decision over the prediction set recovers the optimal VaR-based decision rule.
> >
> > The theme of Kiyani et al. (2025) is highly similar to our work, as we are adopting similar tools (e.g., conformal prediction, optimizations) to analyze decision-making under uncertainty.
> > However, some technical and objective differences separate them.
> > First, conformal prediction is used differently in both works: Kiyani et al. (2025) use conformal prediction sets to construct the uncertainty set of an RO, while our work uses conformal prediction to trace the coverage probability of an inverse feasible region that is from a generic constrained optimization. Second, Kiyani et al. (2025) is a prescriptive framework, while ours is descriptive---instead of outputting an optimal decision, our algorithm aims to audit given decisions by telling how likely they are optimal. Therefore, both works contribute to the study of algorithms for guiding decision-making from complementary aspects.
> >
> > ## Q1: Noisy trend in Figure 5 Column 2
> >
> > The presented "noisy" trend of the point model is due to the randomness of the data simulation procedure.
> > Across different $K$, we are independently simulation training data and refitting the prediction models.
> > Therefore, the point prediction model will be slightly different across different $K$, which will result in a line that is not completely flat.
> > We note that the noisy trend does not harm the main takeaway of the figure, as it can be seen from the figure that the overall trend is flat (i.e., the mean of the curve does not shift with different $K$).
> > This serves as a contrast to CREDO's curve, demonstrating one of CREDO's core properties---the risk estimate becomes more and more accurate as the number of generation size $K$ increases.
> >
> > ## Q2: Exclusion of NS
> >
> > The main reason we exclude NS in Columns 2 and 3 is to avoid unnecessary confusion.
> > NS is an ablation model created from CREDO that ignores validity but purely optimizes for accuracy. Therefore, NS is naturally expected to perform poorly on validity, but would perform well on the accuracy dimension.
> > This may mislead some readers to believe that NS "works well" as it has high accuracy.
> > However, this is not true, since in our risk-averse problem setting, validity is always prioritized over accuracy. That is, the former is a hard constraint while the latter is a soft constraint.
> > We hope to imply this through our presentation of the experiment and avoid unnecessary confusion, so we eliminate NS in subsequent comparisons of the soft constraints after observing that it is deficient for the hard constraint (i.e., only satisfying validity around 50% of the time).
> >
> > We have also provided the full version of Figure 5 that includes NS in Columns 2 and 3 in our original submission, Appendix Section F, Figure 6-9, to ensure completeness of our results.
> >
> > ## Q3: Potential problem with accuracy
> >
> > We apologize for the confusion and appreciate the reviewer's careful reading. The definition of accuracy contains a typo: in our experiment, accuracy is defined as the *negative* of the absolute difference between true and estimated risk (i.e., -MSE), not just the absolute difference (i.e., MSE).
> > With this corrected definition, a higher accuracy indeed indicates better model performance, and the original interpretation provided in Lines 432-437 is logically correct.
> >
> > We also gently note that our original code for implementing accuracy uses this correct definition, which can be found in the supplementary material, exp1_tradeoff.ipynb, within the "relative_accuracy" function.

---

> > > ### Author Response · Authors · 2025-11-26
> > >
> > > Dear reviewer pzmF,
> > >
> > > We really appreciate the constructive feedback, and we hope our response has satisfactorily addressed your concerns, especially regarding the decision quality metric. As the deadline for the discussion period is closing in, we would like to hear your further thoughts, concerns, or questions regarding our paper. We are always happy to engage in further discussions. Thank you!
> > >
> > > Sincerely,
> > >
> > > Authors

---

### Official Review · Reviewer_TYnS · 2025-11-01

**Soundness:** 3
**Presentation:** 2
**Contribution:** 2
**Rating:** 4
**Confidence:** 4

**Summary:**

The paper introduces a framework that provides distribution-free upper bounds on the probability that a given decision is suboptimal. The authors use inverse optimization space of the outcome and then construct a conformal set that is contained in the inverse space to produce the upper bound on the probability of suboptimal decision. The authors give a computational efficient algorithm when the objective function is a linear combination of decision and outcome. The authors validate the framework by experiments on synthetic and power-grid planning datasets.

**Strengths:**

- The paper's motivation is practical. The interpretable risk assessment in high-stakes domains is widely applicable and important.
- The theorems on conservativeness and the Monte Carlo interpretation demonstrate sound reasoning and attention to statistical guarantees.
- The idea of using inverse optimal space of outcome to find an upper bound on distribution-free probability is refreshing and can lead to potentially stronger results.

**Weaknesses:**

- The computational efficiency of the framework is not discussed. Finding the inverse space of the outcome where a given decision is optimal can be NP-hard for any objective function. It is also NP-hard to check whether the conformal set is included by the inverse space. The radius assumption here is still not enough since the inverse space can be non-convex.
- The upper bound that is found by the framework could be arbitrarily bad. That says there is no result on the lower bound of the probability of decision being suboptimal. I'm having this worry especially because the conformal set algorithm constructs the candidate space as a naive radius space. It is very easy to construct a case where the radius is arbitrarily small such that $\alpha$ is arbitrarily large.
- The linear form of objective function is not a very general form. A lot of utility functions in decision-making such as brier score cannot be converted to linear form.
- I found the paper is a little hard to follow. Some properties are not discussed. See more details in questions.

**Questions:**

- What are the intuitions on the generative model $\hat{f}$? How does it impact the quality of $\alpha$? Is there any guideline for choosing $\hat{f}$?
- Why the radius version of conformal set is taken instead of the more general one?
- What does this repeat K times do? What is the random variable here?

---

> ### Author Response · Authors · 2025-11-19
>
> We sincerely appreciate Reviewer TYnS’s thoughtful compliments on the strength of this work. We are encouraged by the reviewer’s confidence in the potential of CREDO and in the soundness of its methodological foundations.
> We are also sorry to hear that parts of the paper were difficult to follow. In this response, we do our best to resolve the reviewer's questions, and we are more than happy to engage in further discussions.
>
> ## W1: NP-hard
>
> We apologize for the confusion regarding the computational aspect of this work. While it is indeed an NP-hard procedure to determine set inclusions, for the convex setting, we have found that there exist alternative computational reformulations of the algorithm where NP-hard procedures can be fully avoided.
> This can be seen by integrating the Monte Carlo interpretation (Proposition 2) and closed-form solution (Corollary 1):
>
> > Under the convex setting, the proposed risk estimator can be equivalently written as:
> > $$
> > \hat{\alpha}(z) = 1 - \frac{1}{K} \sum_{k = 1}^K \left( w^{(k)}(x, z) \cdot \mathbf{1} [ z \in \pi( \hat y^{(k)}) ] \right),
> > $$
> > where $\mathbf{1}[\cdot]$ is the indicator function, and $w^{(k)}$ is the conformalized weight, defined as
> > $$
> > w^{(k)}(x, z) = 1 - R^{-1} \left( V \right),
> > $$
> > and $V$ is the objective value to the following optimization problem:
> > $$
> > \min_{y, z'} \| y - \hat y^{(k)} \|_2 \quad \text{s.t.} \quad y \in \mathcal{Y}, ~ z' \in \mathcal{Z}, ~ g(z; y) - g(z'; y) > 0.
> > $$
>
> Therefore, to estimate this reformulated risk estimator, no NP-hard procedure needs to be invoked, as both $\pi$ and the last optimization problem (which is an SOCP) can be solved by standard off-the-shelf convex solvers such as Gurobi.
> This result further augments the linear result we've derived in the paper to demonstrate the computational efficiency of the algorithm.
>
> Though, as the reviewer suggested, we acknowledge that a discussion on computational efficiency is necessary, as the computation cost is nontrivial for a convex setting due to the need for solving optimizations.
> We will include a detailed discussion of the computational complexity of our approaches in the revised paper.  Additionally, we acknowledge that the extension to arbitrary (e.g., nonconvex) objective functions is challenging and is unclear at the moment.
> We will also include this as a discussion in the limitations section of our revised paper.
>
> ## W2: lower bound on risk
>
> We acknowledge that the current paper doesn't include a detailed lower bound ("tightness") theoretical guarantee for the estimated upper bound.
> However, we wish to note that our experiment results show that the CREDO model produces risk estimates that are empirically tighter compared to naive baseline methods (Figure 5 Col 2-3) and also tight enough to be informative to make high-quality decisions (Table 1).
> These results support that CREDO can provide nontrivial risk estimations that are useful in assisting decision-making, which is the ultimate goal of this framework.
>
> We would also like to note that the key ingredient that we introduced to enable such improvement is the generative model. It produces randomized generations of the centroid of the balls (i.e., radius spaces) within the outcome space.
> So, while it is true that the reviewer noted that one could construct a case where the radius is arbitrarily small, this only happens with low probability, as it only occurs when the generated centroid is near the boundary of the inverse feasible region.
> Therefore, it is unlikely that the average of these estimated risks will degenerate.
> A more in-depth discussion on generative modeling is included in our response "Q1: Role of generative model".
>
> ## W3: linear assumption
>
> We fully agree with the reviewer that the linear assumption is limiting in practice.
> Though we hope to clarify that the purpose of Corollary 1 is to serve as a case-study of CREDO to illustrate its intuitions and show its nice property of having a closed-form solution.
> It is not always required for CREDO to be implemented in practice and/or general settings, but it would be hard (if not impossible) to derive without linear assumption.
> We will incorporate a discussion of the limitations of the linear assumption in practice.

---

> > ### Author Response · Authors · 2025-11-19
> >
> > ## Q1: Role of generative model
> >
> > Intuitively, the role of the generative model is to help mitigate the overconservatism in the risk estimate.
> > Note that a nontrivial risk certification requires the conformal prediction ball with moderate size to be entirely contained within the inverse feasible region.
> > Using generative models enables multiple stochastic draws, increasing the likelihood that at least one sample lies within the inverse feasible region with sufficient surrounding space.
> > Otherwise, using point prediction models may produce outputs that lie near the boundary of, or even outside, the inverse feasible region, leading to overly conservative risk estimates.
> >
> > Another more fundamental intuition can be derived from the Monte Carlo interpretation of CREDO.
> > Specifically, we have shown in the paper that the proposed risk estimator can be equivalently written in the following form:
> > $$
> > 1 - \frac{1}{K} \sum_{k = 1}^K w_i \cdot \mathbf{1} [ \hat y^{(k)} \in \pi^{-1}(z) ], \quad \hat y^{(k)} \sim \hat f,
> > $$
> > where the weight $w_k$ is computed by the proposed conformalized procedure to satisfy validity.
> > Therefore, the role of the generative model $\hat f$ is to directly serve as the sampling model to construct such a weighted sample average estimator.
> >
> > This also explains how $\hat f$ impacts the quality of $\alpha$.
> > Namely, having a well-fitted generative model and increasing the sampling size $K$ enhances the accuracy of the estimate.
> > Under these two assumptions, one can manually set the weights $w_k$ to one for maximum accuracy (tightness), as supported by the law of large numbers.
> > However, we note that the validity of the risk estimate (i.e., being the upper bound) is irrelevant to how well the generative model is fitted and the number of samples $K$, as the proof for Theorem 1 is inherently agnostic to these quantities.
> >
> > There are no specific requirements for $\hat f$ except for being a good approximator for $P(Y|X)$ and can be efficiently sampled from.
> > Empirically, this would require that the model be specified with enough expressivity without too much redundancy. Our experiment adopts Gaussian Mixture Models as the generative model, as it suffices to model low-dimensional cases (e.g., $2$D multimodal distributions). More advanced models, such as VAE and/or diffusion models, can be used in higher-dimensional cases.
> >
> > ## Q2: Choice of conformalized radius
> >
> > The e-value conformalized radius is exclusively used in our paper because it guarantees post-hoc validity, while the standard conformalized radius (a.k.a., the general version) doesn't.
> > This property is crucial for establishing an exact bound for conservatism (Theorem 1) in our paper and ensuring the theoretical soundness of the proposed approach.
> >
> > In fact, when using the standard conformal prediction radius, the bound in Theorem 1 will be contaminated with a nontrivial error term, as stated in the following theorem:
> >
> > > Under the exchangeability assumption and setting $R$ to the standard conformal prediction radius, the estimator $\hat{\alpha}(z)$ satisfies:
> > > $$
> > > \mathbb{P} [ z \in \pi(Y) ] \geq 1 - \mathbb{E} \left[ \hat{\alpha}(z) \right] - \frac{\sum_{i = 1}^n  d_{\rm TV}^{(i)}(\hat \alpha(z)) }{n+1},
> > > $$
> > > where $d_{\mathrm{TV}}^{(i)}(\hat{\alpha}(z))$ denotes the total variation distance between the conditional distribution of the data vector $\{(X_i, Y_i) \}_{i = 1}^n \cup \{(X, Y)\}$ and the same vector with its $i$-th and $(n+1)$-th entries swapped, given the $\sigma$-field generated by $\hat{\alpha}^{(k)}(z)$.
> >
> > This result contrasts with the tight bound shown in Theorem 1, implying how e-value conformalized radius serves as a crucial component in supporting the safe deployment of CREDO.
> > The proof of the statement above directly follows by combining Theorem 2 of Barber et al. (2020) and the proof technique of Theorem 1 in this paper.
> >
> > ## Q3: Definition of $K$ and the random variable
> >
> > The hyperparameter $K$ denotes the number of Monte Carlo samples, whose intuition is detailed in our previous response to "Q1: Role of generative model". Increasing $K$ improves the accuracy of this approximation by reducing Monte Carlo variance, but potentially comes at a higher computational cost as it requires solving more instances of the conformalized procedure.
> > The random variable in this context is the outcome variable $Y$, which captures the uncertainty inherent in a decision-making-under-uncertainty problem.
> >
> > For example, we find in our experiment that setting $K = 100$ would guarantee a sufficiently tight risk estimate. In our grid planning experiment, $Y$ is defined as the number of adopted solar panels (in number of units) in the next year of the service territory.

---

> > > ### Author Response · Authors · 2025-11-26
> > >
> > > Dear reviewer TYnS,
> > >
> > > We really appreciate your helpful comments, and we hope our response has satisfactorily addressed your questions. As the deadline for the discussion period is approaching, we would like to know if you have any follow-up questions or concerns. We are always happy to engage in further discussions to help clarify potential ambiguities. Thank you!
> > >
> > > Sincerely,
> > >
> > > Authors

---

### Official Review · Reviewer_XFwE · 2025-11-01

**Soundness:** 3
**Presentation:** 2
**Contribution:** 3
**Rating:** 6
**Confidence:** 4

**Summary:**

CREDO provides a distribution-free upper bound on the probability that a candidate decision is suboptimal, using inverse optimization geometry and conformal prediction with generative models. It enables practitioners to audit both algorithmic and expert-proposed decisions, offering statistically valid risk certificates. Theoretical guarantees and empirical results on synthetic and real-world tasks demonstrate that CREDO delivers conservative, interpretable, and actionable risk estimates, improving trust and decision quality compared to standard PTO and robust optimization approaches.

**Strengths:**

1. The paper addresses an under explored research area about how to provide rigorous, interpretable risk certificates for candidate decisions in high stakes, uncertain environments.
2. The "decide-then-assess" paradigm is a reasonable variation from the standard "predict-then-optimize" pipeline, and is well-motivated by practical needs for human-AI collaboration.
3. The use of inverse optimization geometry to characterize the optimality region for a decision is well done.
4. The integration of conformal prediction with generative models for risk estimation and the corresponding theoretical guarantees are clearly stated and proved.
5. The closed form solution for linear programs makes the method practical for large scale problems.
The experiments are well designed, covering both synthetic and real world settings (e.g., power grid planning).

**Weaknesses:**

1. While the method is general, the closed form efficiency is only for linear programs. For nonlinear or combinatorial problems, the computational cost of characterizing the inverse feasible region may be significant.
2. The approach assumes access to a well calibrated conditional model for the uncertain parameters. While the paper uses generative models to estimate the conditional distribution, in practice, any model that can accurately capture and sample from P(Y∣X) would suffice, including parametric or non-parametric approaches. I think it is limiting to claim the importance of generative models in this use case.
3. The method is conservative by design, but this can lead to loose risk estimates in some settings. The paper discusses this tradeoff, but more empirical analysis of the "tightness" of the certificates would strengthen the work.
4. The paper is motivated by human AI collaboration, but there is little discussion or experimentation on how practitioners actually use or interpret the risk certificates. A user study or qualitative feedback would be valuable.
5. The paper positions itself relative to robust optimization, DRO, and conformal prediction, but could more deeply discuss how CREDO compares to recent advances in human-in-the-loop optimization.

**Questions:**

1. For general nonlinear or combinatorial optimization problems, how is the inverse feasible region (\pi^{-1}(z;\theta)) practically characterized? Are there efficient relaxations that maintain the validity of the risk certificate, or does the method require exact computation?
2. The framework uses conformal prediction with generative models to construct inner approximations of the inverse feasible region. How sensitive is the risk estimate to the choice of conformal set (e.g., L2 balls vs. other shapes)?
3. What are the theoretical or empirical sample complexity requirements for the calibration set to ensure valid and non-trivial risk certificates, especially as the dimension of Y increases? How does the method perform with limited calibration data?
4. Can the CREDO framework be extended to settings where decisions are made sequentially or in multiple stages, with uncertainty revealed over time? What are the main challenges or limitations in such extensions?
5. Beyond generative models, have the authors empirically compared CREDO with approaches using quantile regression, Bayesian models, or ensemble methods for conditional uncertainty estimation?
6. Can the authors provide a more detailed analysis of the computational complexity of CREDO for both the linear and general cases, including the cost of generating samples, constructing conformal sets, and evaluating the inverse feasible region?

---

> ### Author Response · Authors · 2025-11-19
>
> We really appreciate Reviewer XFwE's positive comments on the strength of this work and providing a constructive review of the paper. We hope this response can adequately address the questions raised by the reviewer, and if not, we are more than happy to engage in further discussions.
>
> ## Responses to weaknesses
>
> We appreciate the reviewer’s constructive comments regarding the weaknesses of our paper. As the points raised are clear and direct, we provide relatively concise responses below.
>
> ### W1
>
> We agree with the reviewer that the closed-form solution is available only for linear programs, and computing inverse feasible regions for general settings can be costly. However, for convex optimization (potentially nonlinear and combinatorial), we have found that there exist alternative computational reformulations of the algorithm where computation of the inverse feasible region can be fully avoided. This can be seen by integrating the Monte Carlo interpretation (Proposition 2) and the closed-form solution (Corollary 1):
>
> > Under the convex setting, the proposed risk estimator can be equivalently written as:
> > $$
> > \hat{\alpha}(z) = 1 - \frac{1}{K} \sum_{k = 1}^K \left( w^{(k)}(x, z) \cdot \mathbf{1} [ z \in \pi( \hat y^{(k)}) ] \right),
> > $$
> > where $\mathbf{1}[\cdot]$ is the indicator function, and $w^{(k)}$ is the conformalized weight, defined as
> > $$
> > w^{(k)}(x, z) = 1 - R^{-1} \left( V \right),
> > $$
> > and $V$ is the objective value to the following optimization problem:
> > $$
> > \min_{y, z'} \| y - \hat y^{(k)} \|_2 \quad \text{s.t.} \quad y \in \mathcal{Y}, ~ z' \in \mathcal{Z}, ~ g(z; y) - g(z'; y) > 0.
> > $$
>
> Therefore, to estimate this reformulated risk estimator, no inverse feasible region needs to be estimated, as both $\pi$ and the last optimization problem (which is an SOCP) can be solved by standard off-the-shelf convex solvers such as Gurobi.
> This result further augments the linear result we've derived in the paper to demonstrate the computational efficiency of the algorithm.
>
> ### W2
>
> We apologize for the potential confusion, and we appreciate the reviewer's clarification. By generative model, we are indeed referring to statistical models that can characterize a conditional distribution and allow sampling from it. This is to be distinguished from deep generative models used to generate images, text, and other high-dimensional content. We will clarify this distinction explicitly in the revised paper.
>
> ### W3
>
> We agree with the reviewer that we do not have a theoretical characterization of the tightness of the risk estimates.
> Though we gently note that our experimental results include an empirical assessment of it. Specifically, Figure 5 (Columns 2–3) shows that CREDO produces substantially tighter risk estimates than the naive baselines as the number of simulation samples and the underlying data variance increase. Figures 10, 11, and 16 in the appendix also explicitly visualize the estimated risk values as well as their "tightness". This is to show that our intention aligns well with the reviewer on the importance of maintaining a valid yet tight risk estimate.
>
> ### W4
>
> We fully agree with the reviewer that a real-world user study with qualitative feedback under the CREDO framework would strengthen our argument on its practical value in human-AI collaboration. We are currently actively looking for collaboration and funding opportunities to conduct such a user-interaction (human-in-the-loop) experiment.
>
> ### W5
>
> We appreciate and agree with the suggestion from the reviewer. We will include a discussion of the human-in-the-loop optimization in the related work section of the revised paper.
>
> ## Q1: Computation of inverse feasible region
>
> For nonlinear and combinatorial objective functions, inverse feasible regions generally need to be characterized by a heuristic brute-force algorithm as detailed in Algorithm 2 of the Appendix.
> The approximation precision can be controlled by the user by tuning the discretization resolution hyperparameter of both the outcome space and decision space. A higher discretization resolution leads to a more refined risk estimate, but would also incur higher computation cost (See response "Q6: Computation complexity").
> Though we note that this procedure may be entirely avoided if one adopts the form of estimator in the theorem presented in "W1" under the convex setting.
>
> The validity of the risk certificate relies on the ability to compute the quantity exactly. Consequently, the use of a brute force algorithm and the more principled convex algorithm (unless assuming perfect optimization convergence) may both introduce deviations that could compromise this validity. To the best of our knowledge, under the current framework, exact computation is only tractable in the linear setting, where validity can be maintained rigorously. The reviewer's intuition is correct.

---

> > ### Author Response · Authors · 2025-11-19
> >
> > ## Q2: Shape of conformal set
> >
> > The reviewer's interpretation is correct and raises a very interesting question. Our understanding is that the choice of the shape conformal set will not affect the validity of the method, as the proof for the validity theorem (Theorem 1) is agnostic to it.
> > However, it may affect the accuracy (i.e., tightness) of the risk estimate, as it changes how tightly the risk upper bounds the true coverage level of the conformal set.
> > We hypothesize its sensitivity could depend on multiple factors, such as the number of generated samples $K$, the underlying distribution form, and the shape of the inverse feasible regions.
> >
> > Though we do note that $\ell_2$ ball is superior to other shapes for computation efficiency reasons.
> > In particular, when using other shapes, he closed-form solution in the linear setting no longer holds, as it relies on a classical closed-form expression for the Euclidean distance from a point to a hyperplane, which is unique to $\ell_2$ balls. This result may generalize to all $\ell_p$ balls (e.g., squares) and some nonlinear settings, so it may benefit from starting with $\ell_2$ balls as a default choice in further studies.
> >
> > ## Q3: Sample complexity
> >
> > We appreciate the reviewer's question on the sample complexity of CREDO.
> > Empirically, we find that CREDO can generally produce non-trivial risk estimates even with moderate sample sizes.
> > Empirically, we find that in our synthetic experiment, CREDO demonstrates substantially improved performance over baseline methods using only 100 calibration samples, and we observe that this number can be reduced to roughly 10 without materially affecting the results. In our real-world experiment, CREDO also achieves the highest decision-quality metrics using only about 150 monthly DER-adoption data points. The estimated risk compared with the ground truth is illustrated in Figures 10, 11, and 16 in the appendix, which are all non-trivial, further illustrating its practical efficiency with limited calibration data
> >
> > However, in high-dimensional settings, CREDO's sample efficiency can be greatly hampered.
> > Empirically, we find that CREDO's estimation degrades quickly in higher dimensions and becomes too loose to be useful around 10D, while our current experiment focuses on 2D and 4D settings.
> > A plausible explanation is that, as the dimensionality increases, the conformal set will gradually take up more volume even at the same confidence level, a well-known phenomenon referred to as the curse of dimensionality, which contributes to overly conservative risk estimates.
> > Theoretically characterizing this phenomenon is a highly nontrivial effort, as it involves characterizing the complex interplay between the geometries, data distribution, and the configuration of the generative model, which we leave to future work.
> >
> > ## Q4: Extending to sequential settings
> >
> > Yes, CREDO can be extended to the sequential setting.
> > A natural approach is to break down the sequential problem into separate static decision-making problems. where for each stage, all available historical information -- including past observations and past decisions -- can be treated as the input $X$ to CREDO, which then produces a stage-wise risk estimate. These per-stage estimates collectively form marginalized decision-risk assessments across the sequence.
> > This procedure is also reflected in our real-world application, where we apply CREDO independently for each year from 2010 to 2024 to generate annual risk assessments.
> > The main limitation of this approach is that it does not yield a joint risk assessment over the full planning trajectory, which is typically more informative in practice than isolated stage-wise evaluations. Addressing the latter requires a fundamentally new formulation of sequential decision-risk assessment, which is the main challenge and remains an open question.
> >
> > ## Q5: Additional baseline comparisons
> >
> > We really appreciate the reviewer's questions.
> > Our current paper does not include the three baselines mentioned by the reviewer to compare with CREDO, but we fully agree that the review raises a valid potential experiment that we should conduct.
> > We are currently working on building the baseline models and will update the experiment results in the revised paper when finished.
> > We plan to construct the three baselines as ablation models for the second step of the algorithm, where instead of using the generative conformal prediction approach to construct conservative risk estimates, we directly apply the three models as probability estimators to estimate the set containment objective $\mathbb{P}[ z \in \pi^{-1}(Y) ]$.
> > We appreciate the suggestion from the reviewer.

---

> > > ### Author Response · Authors · 2025-11-19
> > >
> > > ## Q6: Computational complexity
> > >
> > > We appreciate the reviewer’s question on the computational aspects of CREDO. Below, we provide a detailed discussion of the computational cost associated with (i) generating samples, (ii) constructing the conformal sets, and (iii) evaluating the inverse feasible region for potentially different settings.
> > >
> > > The cost of generating samples depends solely on the generative model used. Common generative models are extremely efficient to be sampled from.
> > > For example, a Gaussian Mixture Model first draws a mixture component according to a categorical probability, and then samples from the corresponding Gaussian.
> > > This typically is a $O(n d^2)$ procedure, with $n$ being the number of components in the mixture model, and $d$ is the dimension of data.
> > > More advanced generative models, such as variational autoencoders (VAE), draw latent variables $z$ from the prior (usually standard Gaussian), and then pass $z$ through the neural-network decoder to generate a sample.
> > > The complexity is mainly dominated by the second step with $O(L w^2)$, with $L$ being the number of neural network layers, and $w$ being the width of each layer.
> > > Both GMMs and VAEs provide fast sampling, making them especially attractive when CREDO requires a modest number of Monte Carlo samples.
> > >
> > > The cost of constructing the conformal sets depends solely on the choice of their geometric form. For $\ell_p$-ball conformal set, the construction cost is essentially negligible, since checking membership of any point in the conformal set then reduces to evaluating whether its distance from the prediction is less than the stored conformalized radius.
> > > For a general conformal set shape, the conformal set would be constructed via discretizing the space and then enumerating points within, so the cost scales with $O(m)$, where $m$ is the number of discretizations.
> > > Combining with our response in "Shape of conformal set", it is recommended that the $\ell_2$ ball be chosen for maximal computation efficiency.
> > >
> > > The cost of constructing an inverse feasible region is large compared to the two previous components and differs significantly depending on the optimization structure.
> > > For a linear programming problem, the inverse feasible region is the intersection of a finite number of halfspaces, where the number is the total number of vertices of the feasible region polygon. Therefore, the cost would be $O(|\mathcal{V}| \cdot d)$, where $|\mathcal{V}|$ is the number of vertices, and $d$ is the data dimension that originates from computing linear inequalities of halfspaces.
> > > For a general optimization problem, one would rely on a heuristic algorithm which constructs the inverse feasible region via discretizing the space and then enumerating points within, so the cost scales with $O(m^2 \cdot C_g)$, where $m$ is the number of discretizations, which is doubled since discretization is applied to both the outcome space and the decision space, and $C_g$ is the complexity for solving the optimization problem.
> > > Both constants are substantially larger compared to other defined constants; therefore, for general objective functions, computing the inverse feasible region is the main overhead.
> > >
> > > Though we again note that we have found that in the convex setting, the computation of the inverse feasible region can be avoided when computing risk estimates. This is made possible through the theorem we present in our response to the first weakness in "W1". This makes the algorithm further computationally efficient to implement, without the need to rely on the computational complexity bound above.
> > > The theorem incurs a computational complexity bound of $O(C_{SOCP} + C_g)$ for one iteration of $k$, where $C_{SOCP}$ additionally denotes the computation cost for solving the SOCP, which is much smaller than the multiplicative bound of $O(m^2 \cdot C_g)$ for finding the inverse feasible region, especially with a large $m$ specified by the user.

---

> > > > ### Author Response · Authors · 2025-11-26
> > > >
> > > > Dear reviewer XFwE,
> > > >
> > > > We really appreciate your helpful suggestions, and we hope our response has satisfactorily addressed your questions. As the deadline for the discussion period is just around the corner, we would like to know if you have any further questions or concerns regarding our paper. We are always happy to engage in further discussions. Thank you!
> > > >
> > > > Sincerely,
> > > >
> > > > Authors

---

### Official Review · Reviewer_yHqA · 2025-11-24

**Soundness:** 3
**Presentation:** 3
**Contribution:** 2
**Rating:** 6
**Confidence:** 4

**Summary:**

This paper proposes a method, to provide a statistically valid estimates of the probability of sub-optimality for any candidate decision proposed by a human expert. They do this through rewriting this problem in terms of an inverse optimization problem of the outcome and then construct a conformal set that is contained in the inverse space. They also provide a computationally efficient implfememntation of this method.

**Strengths:**

- The problem of designing ML-powered, distribution-free valid risk certificates is a timely and impactful problem. The authors make a meaningful step by connecting this problem to running CP on an inverse optimization problem.
- Their work has a very good balance of theory and experiment. The theoretical guarantees are of the interest of practice, and they showcase that it actually works in real world datasets.

**Weaknesses:**

My first concern is regarding "selection bias". Selection bias is well-known phenomena is statistics, which points toward the scenario where a decision maker wants to use an estimation to inform their decisions. The bias arises, when the estimation is performed without the knowledge of the down stream decision problem, and this can potentially disrupt the statistical guarantees of the original estimation. In the context of the problem that is studied in this paper, it shows itself as follows: say a decision maker want to pick a decision such that the risk certificates is larger than \tau. Now what they could do is to run your algorithm, get the certificate, and then filter out the decisions that the corresponding certificate is less than \tau, and pick one of the remaining. Now the issue is, from the viewpoint of the decision maker, what actually matters is that the risk certificate be statistically valid conditioned on the certificates larger than \tau. If this doesn't hold, then the algorithm suffers from "selection bias". The question is, whether this algorithm suffers from such a situation (my guess is yes), and if yes, to what extend (this could be certified in experiments), and if the bias is significant, how to fix it (this could at least be discussed\reported in the future works and limitations).

The second concern is regarding a proper discussion regarding the two recent works of [1] and [2]. Although the scope of those papers, particularly [1], is to "find" a low-risk action, rather than "certify" the low risk ones, however, one can use their method to derive risk certificates too. The idea is, you can run your favorite CP method, to get a set of labels C(x). Then if you run the max-min rule defined in these papers, you get both a certificate (as the max-min value), and an action (the argmax-min). Alternatively, you can ignore the outer max, and for each action candidate of your interest, you can just solve the inner min of their method, and that would give you a risk certificate that holds with high probability. There needs to be a discussion to distinguish this simple method with yours, in terms of scope, practicality, and use cases.

I like this paper, and if the authors provide a satisfying answer to these concerns I would be happy to vote for acceptance.


[1]: Decision Theoretic Foundations for Conformal Prediction: Optimal Uncertainty Quantification for Risk-Averse Agents, Kiyani et. al.

[2]: Certified Decisions, Andrews et. al.

**Questions:**

.

---

> ### Author Response · Authors · 2025-11-26
>
> We greatly appreciate the reviewer’s positive reception of our paper. Below, we provide a detailed response to the reviewer's concerns, and we are more than happy to engage in further discussions on the two points raised by the reviewer.
>
> ## Q1: Selection bias
>
> This is a very interesting problem, though we would first like to kindly emphasize that the main focus of the paper is to study the problem of decision risk assessment and to propose an algorithm that can estimate the risk reliably.
> It is beyond the scope of the current paper to study specific decision-making procedures based on the obtained risk estimate and their associated subsequent issues.
> However, we are more than happy to share our thoughts on the reviewer's question and record it in the future work sections of the revised paper.
>
> Now, to formalize the problem: given an estimated risk certificate $\hat \alpha(z)$ and some threshold $\tau \in (0, 1)$, we want to know whether the validity guarantee still holds conditioning on the estimator being thresholded at $\tau$:
> $$
> \mathbb{P}\left[ z \in \pi(Y; \theta) \right] \ge 1 - \mathbb{E} \left[ \hat \alpha(z) \mid \hat \alpha(z) < \tau \right].
> $$
> As the reviewer suggested, this problem could arise in scenarios where the decision-maker filters the decisions by their risks, potentially raising the selection bias issue.
> This is also a nontrivial problem, because the right-hand side is now a larger quantity compared to the original bound $1 - \mathbb{E}[\hat \alpha(z)]$, which could potentially break the guarantee.
>
> This statement is challenging and, to the best of our knowledge, cannot be proved using the theoretical tools developed in this paper.
> For instance, attempting to reuse the proof strategy of Theorem 1 would require relaxing Lemma 1 (e-value post-hoc validity) from a marginal guarantee to a conditional one given the event ${\hat{\alpha}(z) < \tau}$. This appears infeasible, as the bound in Lemma 1 is already tight. Therefore, similar to the reviewer, we believe this guarantee can no longer be established under the selection bias setting.
>
> To address this fundamentally, we believe the algorithm has to be modified.
> One potential direction is to stratify the estimation procedure by filtering out generated samples $\hat \alpha^{(k)}(z)$ with $\hat \alpha(z) \geq \tau$, but would require knowing $\tau$ in advance.
> Another way is to adopt a more conservative conformalized radius design, such as log-transformed e-value statistics, but would usually require additional distributional assumptions about the data (Ramdas and Wang, 2024).
>
> Empirically, we also implemented the selection bias decision-making scheme in our two synthetic settings. The results are shown in the two tables below.
> It can be seen that for both settings, "validity w/ bias" degrades quickly as we decrease the value of $\tau$. This is because decreasing $\tau$ is equivalent to enforcing a stricter selection criterion over the decision risk, and therefore incurs stronger selection bias that disrupts the validity guarantee.
> Looking at "difference of $\hat \alpha$", it can also be seen that as $\tau$ decreases, the risk estimates exhibit increasingly pronounced bias from the risk estimate without the thresholding selection procedure. These results jointly suggest that the selection bias effect empirically exists, the magnitude of the bias is strong, and the decision maker should be aware of it in practice to facilitate trustworthy decision-making.
>
> | Metric \ $\tau$ | $\tau = 0.9$ | $\tau = 0.8$ | $\tau = 0.7$ |
> |-|-|-|-|
> |Val | $0.99 \pm 0.07$ | $0.99 \pm 0.07$ | $0.99 \pm 0.07$ |
> |Val (w/ bias)  | $0.98 \pm 0.09$ | $0.90 \pm 0.00$ | $0.67 \pm 0.00$ |
> | Difference of $\hat \alpha$ | $0.06 \pm 0.00$ | $0.13 \pm 0.00$ | $0.21 \pm 0.00$ |
>
> *Table 1: Results under Setting I*
>
> | Metric \ $\tau$ | $\tau = 0.95$ | $\tau = 0.9$ | $\tau = 0.85$ |
> |-|-|-|-|
> |Val | $0.99 \pm 0.06$ | $0.99 \pm 0.06$ | $0.99 \pm 0.06$ |
> |Val (w/ bias)  | $0.91 \pm 0.12$ | $0.83 \pm 0.07$ | $0.60 \pm 0.03$ |
> | Difference of $\hat \alpha$ | $0.02 \pm 0.00$ | $0.03 \pm 0.00$ | $0.05 \pm 0.00$ |
>
> *Table 2: Results under Setting II*
>
> PS: We make a small correction to the reviewer's hypothesized scenario: the decision maker usually chooses $z$ with $\hat \alpha(z) < \tau$ rather than $\hat \alpha(z) > \tau$, because a smaller risk represents a more desirable decision.
>
> [1] Ramdas, Aaditya, and Wang, Ruodu. "Hypothesis testing with e-values." arXiv preprint arXiv:2410.23614 (2024).

---

> ### Author Response · Authors · 2025-11-26
>
> ## Q2: Related work
>
> We thank the reviewer for this helpful comment. Indeed, although the current manuscript cites the two works, the associated discussion is insufficient. We will expand and clarify this discussion in the revised manuscript.
>
> We would like to gently point out that the modified approach of Kiyani et al. (2025) suggested by the reviewer appears not to yield the risk certificate $\alpha$.
> Specifically, their main objective of Kiyani et al. (2025) is
> $$
> \min_{z \in \mathcal{Z}} \max_{y \in \mathcal{C}(X; \alpha)} g (z ; y).
> $$
> The reviewer suggested first dropping the outer minimization, and then considering solving the inner maximization with $z$ as a prespecified constant.
> However, the solution to this inner problem is the variable $y$, not the risk $\alpha$. The risk $\alpha$ still needs to be prespecified by the decision maker to compute the inner minimization, not "outputted" by this procedure as CREDO (our algorithm) does.
>
> The reason lies in the fundamental distinction of how CP is used in our work and Kiyani et al. (2025).
> In our algorithm, CP is used in a *backward* manner. We adaptively search for the largest CP radius that remains feasible, and then trace back the corresponding risk level $\alpha$ for which this radius is valid. The resulting $\alpha$ is subsequently processed to form the final risk certificate.
> In contrast, Kiyani et al. (2025) employ CP in the conventional *forward* direction: the decision maker specifies a desired risk level $\alpha$, and the method constructs a CP set guaranteed to contain $Y$ with probability at least $1 - \alpha$. Therefore, CREDO uniquely treats $\alpha$ as an output, whereas Kiyani et al. (2025) and other literature that integrates CP with RO treat $\alpha$ as an input. This is also one of the key methodological novelties of CREDO.
>
> Therefore, we view Kiyani et al. (2025) and our work as complementary components of a decision-support framework: their work prescribes risk-averse decisions, while ours audits risk estimates for candidate decisions. Each serves a distinct purpose, and neither can be readily replaced by the other. Below, we provide the extended discussion to be included in our related work section:
>
> > Kiyani et al. (2025) examine the foundational role of uncertainty quantification in decision-making for risk-averse agents. They show that when a decision-maker evaluates decision quality using a risk-averse criterion such as VaR, the natural form of uncertainty quantification is a prediction set rather than a calibrated predictive distribution. Their framework reveals a direct connection with RO: when the uncertainty set is given by a prediction set (e.g., produced by conformal prediction), the miscoverage level of the prediction set corresponds exactly to the decision-maker’s target quantile. In this sense, the coverage level of the conformal set determines the degree of risk-aversion, and the resulting max-min decision over the prediction set recovers the optimal VaR-based decision rule.
>
> > Andrews et al. (2025) develop a theory of “certified decisions” that connects inference (e.g., hypothesis tests, confidence intervals) with downstream decision-making under ambiguity. They introduce P-certificates (and in unbounded-loss settings, E-certificates) which give upper bounds on loss that hold with probability at least $1-\alpha$. They show that for ambiguity-averse decision-makers, one can safely recommend actions together with such certificates. Importantly, they prove that without loss one can restrict attention to decision rules that are minimax over confidence sets (i.e., “as-if” decisions over a set estimate)---paralleling the way robust optimization uses uncertainty sets. In this way, the mis-coverage of a confidence set (or the tail of an e-value) corresponds to the decision-maker’s risk tolerance, and the resulting max-min decision over the set yields a recommendation with a valid loss certificate.
>
> > The theme of Kiyani et al. (2025) is highly similar to our work, as we are adopting similar tools (e.g., CP) to analyze decision-making under uncertainty. However, some technical and objective differences separate them. First, CP is used differently in both works: Kiyani et al. (2025) use CP sets to construct the uncertainty set of an RO, while our work uses the inverse of CP to trace the coverage probability of an inverse feasible region that is from a generic constrained optimization.
> This creates a fundamental difference, where Kiyani et al. (2025) take it as an input, while our work treats it as an output.
> Second, both Kiyani et al. (2025) and Andrews et al. (2025) are prescriptive frameworks, while ours is descriptive---instead of outputting an optimal decision, our algorithm aims to audit given decisions by telling how likely they are optimal. Therefore, both lines of work contribute to the study of algorithms for guiding decision-making from complementary aspects.

---

### Author Response · Authors · 2025-12-03
**Summary of reviews**

We really appreciate the reviewers and ACs for the effort you have put into reviewing our paper.
Since we understand that the discussion period has been closed early and all papers have been assigned new ACs, we would like to provide a brief summary of the perceptions of our paper and our main replies.

**Reviewer yHqA (original score: 6, may consider raising score to 8 based on rebuttal)**
stated that they "like the paper" and are happy to vote for acceptance if the response is satisfactory. The main question they raise is whether CREDO may suffer from selection bias and potential ways to address it.
We confirm the reviewer's conjecture, and we point to challenges in its theoretical analysis and potential modifications of the algorithm that may address the issue. Empirically, we conducted additional experiments to show the existence of selection bias and raised some insights that are worth exploring in future work.
The reviewer also suggested including an extensive discussion on two important related works, which we agree on and have also incorporated into the revised paper.

**Reviewer XFwE (original score: 6)** provided detailed comments across multiple aspects of the paper. We find them to be helpful and constructive, and we have addressed them accordingly. Please refer to our rebuttal for further details.

**Reviewer TYnS (original score: 4)**
is primarily concerned with the computational efficiency of the procedure extending beyond linear settings.
We address this by showing that the procedure can be efficiently generalized to any convex setting with a new optimization reformulation that completely circumvents set enumeration/containment.
This optimization can be solved with standard off-the-shelf convex solvers, making the problem computationally efficient beyond linearity.
Additionally, the reviewer also expressed concern about the risk estimate being easily degenerated, which we believe is a misunderstanding, and clarified how incorporating generative models (being one of the core novelties of our paper) exactly tackles the issue.

**Reviewer pzmF (original score: 4, may consider raising score to >= 6 based on rebuttal)**
mainly requested justification for how the empirical confidence ranking metric indicates decision quality, and is willing to reconsider their score if the justification is strong, or implement a more indicative metric.
We responded from three aspects:
(i) We first clarify a potential confusion by showing that the metric definition is consistent with how existing metrics are defined (i.e., at the population level).
(ii) We point out several unique advantages of the metric, making it indicative and well-suited for evaluating decision quality in our decision risk assessment settings.
(iii) Empirically, we also conducted an additional evaluation experiment using regret as the metric. The results are promising, suggesting that our proposed algorithm can indeed reliably support high-quality decision-making, where "quality" may be defined under different performance criteria.

---

### Meta-Review · Area_Chair_qr6L · 2025-12-22

**Summary:**

This paper aims to give statistically valid estimates of the probability of sub-optimality for decisions considered by a human expert. It builds these estimates through inverse optimization and conformal prediction.

Reviewer concerns that I considered important:
- High risk of selection bias when this method is used in practice
- Computational cost may become burdensome
- Upper bound on risk may be loose and is not theoretically controlled
- No discussion of how to use the method downstream nor user study
- Lack of generality of the objective and reliance on convexity
- Definition of empirical confidence ranking may not be a good metric

**Reviewer Concerns:**

- Selection bias (partially addressed):

The authors agreed this was a valid concern, but one they did not explicitly test because they generally did not test downstream applications. While the possibility of selection bias was more thoroughly discussed, this remains a limitation.
- Cost (addressed):

 For convex cases the method is not burdensome, but a more general case is out of reach.
- Loose bound (partially addressed):

The authors were unable to provide theoretical control with an upper bound, but showed empirical evidence that looseness was not a major concern.
- Downstream uses (outstanding):

The authors positioned downstream uses and a user study as out-of-scope of this work.
- Generality (partially addressed):

The authors admitted convexity as a requirement which can limit the generality of the work, but the implications were discussed.
- Metric (addressed):

The authors gave more complete reasoning behind their choice of metric and justified it by comparison to alternatives in the literature.

Given the interesting topic and quality of the work, I am recommending acceptance. As with any paper, there are directions that could have been explored more completely, but ICLR has page limits that prevent the following of every thread.

**Reviewer Scores:**

yHqA - 6 -> 8 (Reviewer stated they may raise, and their two concerns were discussed fully)

XFwE - 6 -> 6 (Several concerns not fully addressed)

TYnS - 4 -> 4 (Several concerns not fully addressed)

pzmF - 4 -> 6 (Main concerns addressed)

---

### Decision · Program_Chairs · 2026-01-26

Accept (Poster)